# Mitochondrial ROS and HIF-1α signaling mediate synaptic plasticity in the critical period

**Daniel Sobrido-Cameán**[1]*, **Bramwell Coulson**[2], **Michael Miller**[1], **Matthew C. W. Oswald**[1], **Tom Pettini**[1], **David M. D. Bailey**[1], **Richard A. Baines**[2]*, **Matthias Landgraf**[1]*

**1** Department of Zoology, University of Cambridge, Cambridge, United Kingdom, **2** Division of Neuroscience, School of Biological Sciences, Faculty of Biology, Medicine and Health, Manchester Academic Health Science Centre, University of Manchester, Manchester, United Kingdom

* ds918@cam.ac.uk (DS-C); Richard.Baines@manchester.ac.uk (RAB); ml10006@cam.cam.uk (ML)

## Abstract

As developing networks transition from spontaneous irregular to patterned activity, they undergo plastic tuning phases, termed "critical periods"; "critical" because disturbances during these phases can lead to lasting changes in network development and output. Critical periods are common to developing nervous systems, with analogous features shared from insects to mammals, yet the core signaling mechanisms that underlie cellular critical period plasticity have remained elusive. To identify these, we exploited the *Drosophila* larval locomotor network as an advantageous model system. It has a defined critical period and offers unparalleled access to identified network elements, including the neuromuscular junction as a model synapse. We find that manipulations of a single motoneuron or muscle cell during the critical period lead to predictable, and permanent, cell-specific changes. This demonstrates that critical period adjustments occur at a single-cell level. Mechanistically, we identified mitochondrial reactive oxygen species (ROS) as causative. Specifically, we show that ROS produced by Complex-I of the mitochondrial electron transport chain, generated by the reverse flow of electrons, is necessary and instructive for critical period-regulated plasticity. Downstream of ROS, we identified the *Drosophila* homologue of hypoxia-inducible factor (HIF-1α), as required for transducing the mitochondrial ROS signal to the nucleus. This signaling axis is also sufficient to cell autonomously specify changes in neuronal properties and animal behavior but, again, only when activated during the embryonic critical period. Thus, we have identified specific mitochondrial ROS and HIF-1α as primary signals that mediate critical period plasticity.

## Introduction

The emergence of network function is arguably among the least well-understood aspects of nervous system development. It is well known that the emergence of

**Data availability statement:** All relevant data are within the paper and its Supporting information files.

**Funding:** D.S.-C. was supported by the European Molecular Biology Organization (EMBO) with a long-term EMBO fellowship (ALTF 62-2021). This work was supported by a Joint Wellcome Trust Investigator Award to R.A.B. and M.L. (217099/Z/19/Z) and funding from the Biotechnology and Biological Sciences Research Council (BBSRC) to M.L. (BB/V014943/1). The sponsors or funders had no role in the study design, data collection and analysis, decision to publish, or preparation of the manuscript.

**Competing interests:** The authors have declared that no competing interests exist.

**Abbreviations:** ALH, after larval hatching; AOX, alternative oxidase; DA1, dorsal acute 1; DHODH, dihydroorotate dehydrogenase; ETC, electron transport chain; HIF-1α, hypoxia-inducible factor; Ndi1, NADH dehydrogenase 1; NMJ, neuromuscular junction; Prx3, Perexiredoxin-3; RET, reverse electron transport; ROS, reactive oxygen species; VHL von Hippel-Lindau.

network function, which occurs during late stages of nervous system development, correlates with a phase of adjustment during which networks transition from seemingly irregular to more patterned activity. These phases have been termed 'sensitive' or 'critical periods' and are commonly viewed as periods of heightened plasticity. Importantly, they are also highly sensitive to perturbations: errors induced during the critical period generally have lasting impact, with subsequent plasticity mechanisms unable to correct them [1–3]. Thus, "decisions" made during the critical period can specify neuronal properties over the long-term, and abnormal critical period experiences likely contribute to epilepsy and other neurodevelopmental conditions such as schizophrenia and autism spectrum disorder [4,5]. Though critical periods have been studied for decades, notably in mammalian sensory systems [3,6,7], many questions remain. Here, we focused on two fundamental unresolved questions: whether critical periods are principally a network phenomenon, or a property of individual cells and, secondly, the identity of the primary signals directing critical period plasticity.

A difficulty in understanding critical periods has been the complexity of mammalian circuitry, including the realization that different cell types can behave differently [6]. Therefore, we exploit a much simpler experimental system, the fruit fly, *Drosophila melanogaster*, which has enabled the identification of numerous genes and mechanisms that orchestrate the development and function of the nervous system [8–10]. Importantly, the larval locomotor network has a well-defined critical period in late embryogenesis, with features that suggest it is analogous to mammalian critical periods [11,12]. As is the case for mammals, *Drosophila* critical periods, both in the embryo and during pupal metamorphosis, are developmental windows that are fundamentally activity-regulated, with an important role for shaping the network inhibition:excitation balance [11–14]. For example, in *Drosophila,* transient correction of the network inhibition:excitation balance during the embryonic critical period effects a sustained rescue of seizure phenotypes otherwise caused by mutation of the single sodium channel [11–14]. Experimentally, this model system allows reliable access to identified cells of known connectivity and function. Unlike mammals, *Drosophila melanogaster,* as an insect, is not warm-blooded, but poikilothermic, and is therefore vulnerable to external temperature variation. For the larval locomotor network, we found that increases in ambient temperature elevate network activity and, when applied during the embryonic critical period, phenocopy pharmacological and genetic activity manipulations. Thus, heat stress of 32°C, which is ecologically relevant to temperatures that *Drosophila* experience in the wild [15], represents an ecologically relevant stimulus with which to study critical period biology in this system.

Temperature affects biochemical reactions in general, and metabolic and mitochondrial processes in particular [16]. In *Drosophila,* heat stress causes a reversal of the flow of electrons within the mitochondrial electron transport chain (ETC), from Complex-II back to Complex-I. This so-called reverse electron transport (RET) leads to the production of reactive oxygen species (ROS) at Complex-I, due to a partial reduction of oxygen and the formation of superoxide anions. In *Drosophila*, as well as mouse, RET-generated ROS have been linked to improved stress resistance and life span [17–21]. Here, we identified RET-based generation of ROS at mitochondrial

Complex-I as a primary critical period signal that is necessary and sufficient for instructing changes in subsequent network development.

Downstream of mitochondrial ROS, we further identified the conserved Hypoxia-inducible factor 1 alpha (HIF-1α), of which *Drosophila* has only a single homologue, called *similar (sima)* [22–24]. HIF-1α is a sensor for both oxygen and ROS levels, and is a regulator of metabolism [25]. During normal development, HIF-1α regulates many processes, including neurogenesis, angiogenesis, and cell survival [26–28], and its dysregulation is linked to a range of cancers [29]. In summary, we report that critical period plasticity is principally enacted cell autonomously; with metabolic mitochondrial ROS as a primary signal that is transduced by the conserved HIF-1α pathway, instructing subsequent development of cellular excitable and synaptic properties, and thus network stability and behavioral output.

## Results

### Mitochondrial ROS are plasticity signals during the embryonic critical period

The primary signals and pathways that underlie critical period plasticity remain elusive. To identify these, we used the *Drosophila* larval neuromuscular junction (NMJ) as a proven experimental model [8–10]. As a means to induce a robust phenotype, we used transient exposure to 32°C heat stress during a defined embryonic critical period (occurring at 17–19 h after egg laying at 25°C). We have previously shown this manipulation to increase network excitation (i.e., increased action potential firing) phenocopying the effects of direct activity manipulations [30]. Specifically, embryos experiencing 32°C heat stress during their critical period show subsequent changes to larval development of the NMJ. This change is seemingly permanent and manifests as presynaptic terminal overgrowth (Fig 1A–1C; S1 Data) and a decrease in the postsynaptic high-conductance glutamate receptor subunit, GluRIIA (Fig 1D; S1 Data), but not the lower conductance subunit, GluRIIB (Fig 1E; S1 Data).

The NMJ overgrowth phenotype mirrors the outcome following neuronal overactivation induced by ROS signaling, which we and others have previously studied [30–32]. We therefore asked if mitochondrial ROS might be involved, and measured changes during the critical period following exposure to 32°C heat stress. Specifically, we expressed the ratiometric ROS sensor mito-roGFP2::Tsa2ΔCPΔCR, reported to have improved sensitivity and specificity over previously available tools [33] in all muscles using *Mef2-GAL4*. Indeed, quantification showed a significant increase in mitochondrial ROS in embryonic muscles following 32°C heat stress as compared to 25°C controls (Fig 1A and 1B; S1 Data).

To test if mitochondrial ROS were instructive critical period signals, we genetically dampened mitochondrial ROS levels in different ways, asking if this would suppress the critical period heat stress-induced NMJ phenotypes. We limited these genetic ROS manipulations to embryonic stages only, and, spatially, to a single muscle per half segment: the most dorsal muscle, dorsal acute 1 (DA1). Mis-expression of the mitochondria-targeted ROS scavengers, mito-Catalase or Perexiredoxin-3 (Prx3), in muscle DA1, cell-selectively rescued NMJ overgrowth (Fig 1D; S1 Data) and GluRIIA phenotypes that otherwise result from a critical period 32°C heat stress (Fig 1E; S1 Data). At the control temperature of 25°C, these same genetic manipulations show no such effects (Fig 1).

Mitochondria can generate ROS through various means, notably at Complex-I and Complex-III of the ETC. Heat stress has been shown to cause ROS generation at Complex-I in adult *Drosophila*, via reverse flow of electrons from Complex-II back to Complex-I, (termed 'reverse electron transport', or 'RET' [19]). To test if 32°C heat stress in embryos also causes RET-generated ROS in mitochondria, we cell-selectively knocked down a subunit of Complex-I, ND-75, required for RET-generated ROS. To complement this, in another manipulation we neutralized RET-generated ROS by mis-expression of alternative oxidase (AOX), an enzyme found in the inner mitochondrial membrane of many organisms, though not vertebrates or insects [34–37]. In mitochondria, AOX acts as a bypass that transfers electrons from ubiquinone to oxygen, independent of Complexes-I and -III, thereby reducing the probability of electron backflow and subsequent ROS production by RET [19,38–40]. When we targeted either of these genetic manipulations to muscle DA1 during embryogenesis, both rescued the critical period heat stress phenotype, as per NMJ bouton number and GluRII composition (Fig 1), as well

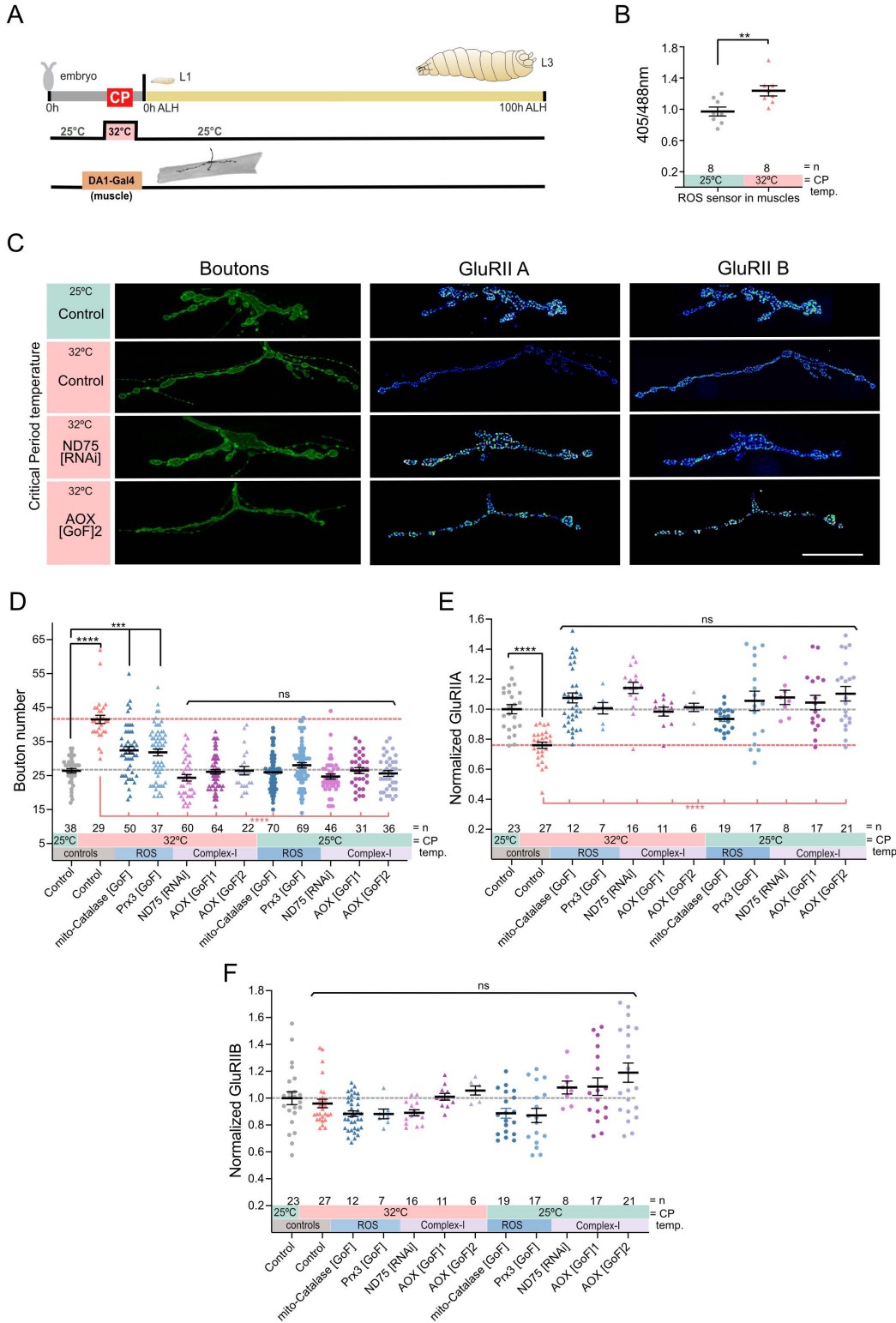

**Fig 1. Mitochondrial ROS generated by reverse electron transport in muscles is necessary for critical period heat stress to change NMJ development.** **(A)** Experimental paradigm. **(B)** 32°C during the critical period increases mitochondrial ROS production in muscles. Dot plots of mitochondrion-targeted ratiometric mito::roGFP2::Tsa2ΔCPΔCR ROS sensor in late stages at 25°C control temperature and 32°C. **(C)** Heat stress experienced during the embryonic critical period (32°C vs. 25°C control) leads to increased aCC NMJ terminal size and decreased postsynaptic GluRIIA,

while not affecting subunit GluRIIB expression. Simultaneous genetic manipulation of muscle DA1 during embryonic stages only identifies mitochondrial ROS generated by reverse electron transport as necessary signals. "Control" indicates control genotype heterozygous for Oregon-R and DA1-GAL4. Larvae were reared at the control temperature of 25°C until the late wandering stage, 100 h after larval hatching (ALH). GluRIIA and GluRIIB subunits are displayed with lookup table "fire" to illustrate signal intensities (warmer colors indicating greater signal intensities). Scale bar = 20 μm. **(D)** Dot-plot quantification shows changes to aCC NMJ growth on its target muscle DA1, based on the standard measure of the number of boutons (swellings containing multiple presynaptic release sites/active zones). Data are shown with mean ± SEM, ANOVA, \*\*\*$p < 0.0001$, \*\*\*\*$p < 0.00001$, 'ns' indicates statistical non-significance. Black asterisks indicate comparison with the control condition of 25°C throughout, genetically unmanipulated. Red asterisks indicate comparisons with control genotype exposed to 32°C heat stress during the embryonic critical period. **(E)** Dot-plot quantification shows changes in levels of the GluRIIA receptor subunit at aCC NMJs quantified in (C). Data are shown with mean ± SEM, ANOVA, \*\*\*\*$p < 0.00001$, 'ns' indicates statistical non-significance. Black asterisks indicate comparison with the control condition of 25°C throughout, genetically unmanipulated. Red asterisks indicate comparisons with control genotype exposed to 32°C heat stress during the embryonic critical period. **(F)** Dot-plot quantification as in (D), but for the low conductance GluRIIB receptor subunit, which remains unaffected by these manipulations. See raw data in S1 Data.

as presynaptic active zone number (S2 Fig; S7 Data). Under control conditions of 25°C these manipulations were again neutral, leaving NMJ development unaffected (Figs 1 and S1; S6 Data). Together, our results show that during the embryonic critical period, RET-generated ROS at Complex-I are necessary for heat stress to effect a lasting change to NMJ development.

We next asked if ROS generated by Complex-I might also be sufficient to induce critical period plasticity in the absence of systemic heat stress. To test this, we cell-selectively induced RET-ROS generation at Complex-I by mis-expression of yeast NADH dehydrogenase 1 (Ndi1). Ndi1 transfers electrons from NADH to the CoQ pool. This can lead to the accumulation of reduced CoQ and transfer of electrons to Complex-I, causing RET and associated ROS generation [28]. Mis-expression of Ndi1in *Drosophila* has previously been shown to increase ROS production in a way indistinguishable from RET [20]. We found that transient embryonic mis-expression of Ndi1in muscle DA1, at the control temperature of 25°C, cell-selectively phenocopies the characteristic critical period heat stress phenotype of NMJ overgrowth (Figs 2A–2C and S1; S2 and S6 Data), reduced GluRIIA levels (Figs 2E and S1) and an increase in active zone number (S2 Fig; S7 Data). Importantly, RET inhibition via AOX suppresses the effects of heat stress (Fig 1), while Ndi1-induced ROS at 25°C phenocopies the 32°C phenotype. To determine whether these manipulations act through a shared pathway, we combined Ndi1 mis-expression with critical period heat stress. This did not further enhance the phenotype, suggesting that temperature and Complex-I-derived ROS converge on the same signaling mechanism (Fig 2). Thus, our results show that RET-generated ROS, produced in mitochondria during the embryonic critical period, are both necessary and sufficient for inducing lasting change to NMJ development.

## Mitochondrial RET-generated ROS are critical period signals for NMJ development

Synapse development requires close interactions between pre- and postsynaptic partners. Therefore, we tested the roles of mitochondrial ROS in a presynaptic motoneuron, termed 'aCC', whose NMJ forms on muscle DA1. We identified a transgenic line that transiently targets GAL4 expression to the aCC presynaptic motoneuron, here referred to as "aCC-GAL4[1]" (aka *RN2-O-GAL4*). Conveniently for critical period manipulations, GAL4 expression by this transgenic line is transient, limited to embryonic stages only, including the critical period. Using this aCC-GAL4[1] to express the mito-roGFP2::Tsa2ΔCPΔCR ROS sensor, we found that motoneurons, like muscles, increase ROS levels in mitochondria following 32°C heat stress during the critical period (Fig 3A and 3B). Next, we tested if ROS is also instructive signals in motoneurons during the critical period, as in muscles. We found that mis-expression of AOX in aCC motoneurons, to neutralize RET-induced ROS generation at Complex-I, was sufficient to rescue the NMJ overgrowth that normally results from embryonic 32°C heat stress (Fig 3A–3D and 3F; S3 Data). Critical period-induced increases in NMJ size (as measured by either bouton number or area) led to a proportionate increase in the number of presynaptic active zones, the sites of vesicular neurotransmitter release, without affecting their overall density (Fig 3E and 3G; S3 Data). However, simultaneous embryonic mis-expression of AOX in the aCC motoneurons fully rescues active zone number (Fig 3D; S3

PLOS Biology

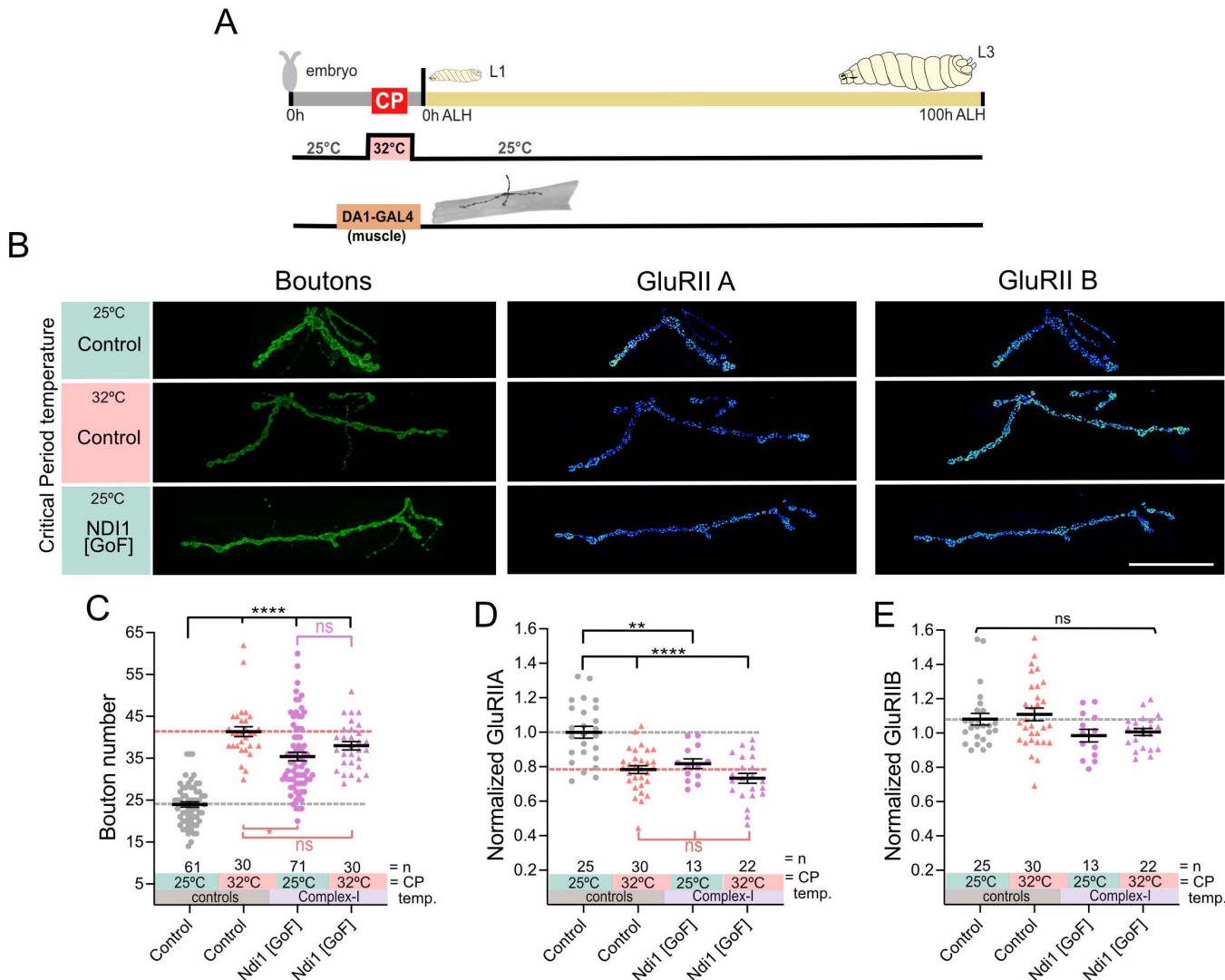

**Fig 2. ROS by Complex-I, during the critical period, is sufficient to induce lasting changes to NMJ development. (A)** Experimental paradigm. **(B)** Induction of ROS at Complex-I via transient mis-expression of NDI1 in muscle DA1 during embryonic stages only phenocopies the effects that critical period heat stress has on subsequent NMJ development. "Control" indicates control genotype heterozygous for Oregon-R and *DA1-GAL4*. Larvae were reared at the control temperature of 25°C until the late wandering stage, 100 h ALH. GluRIIA and GluRIIB subunits are displayed with lookup table "fire" to illustrate signal intensities (warmer colors indicating greater signal intensities). Scale bar = 20 μm. **(C)** Dot-plot quantification shows changes to aCC NMJ growth on its target muscle DA1, based on the standard measure of the number of boutons (swellings containing multiple presynaptic release sites/active zones). Data are shown with mean ± SEM, ANOVA, ***$p < 0.0001$, ****$p < 0.00001$. Black asterisks indicate comparison with the control condition of 25°C throughout, genetically unmanipulated. Red asterisks indicate comparisons with control genotype exposed to 32°C heat stress during the embryonic critical period. **(D)** Dot-plot quantification shows changes in levels of the GluRIIA receptor subunit at aCC NMJs quantified in C). Data are shown with mean ± SEM, ANOVA, **$p < 0.001$, ***$p < 0.0001$, 'ns' indicates statistical non-significance. Black asterisks indicate comparison with the control condition of 25°C throughout, genetically unmanipulated. Red asterisks indicate comparisons with control genotype exposed to 32°C heat stress during the embryonic critical period. **(E)** Dot-plot quantification as in (D), but for the low conductance GluRIIB receptor subunit, which remains unaffected by these manipulations. See raw data in S2 Data.

Data). Conversely, transient embryonic generation of RET-ROS in the aCC motoneurons, by mis-expression of Ndi1, was sufficient to cell-selectively mimic the critical period heat stress phenotype (Fig 3C–3E; S3 Data). This phenotype was not exacerbated by simultaneous exposure to 32°C during the critical period (Fig 3). These findings demonstrate that NMJ

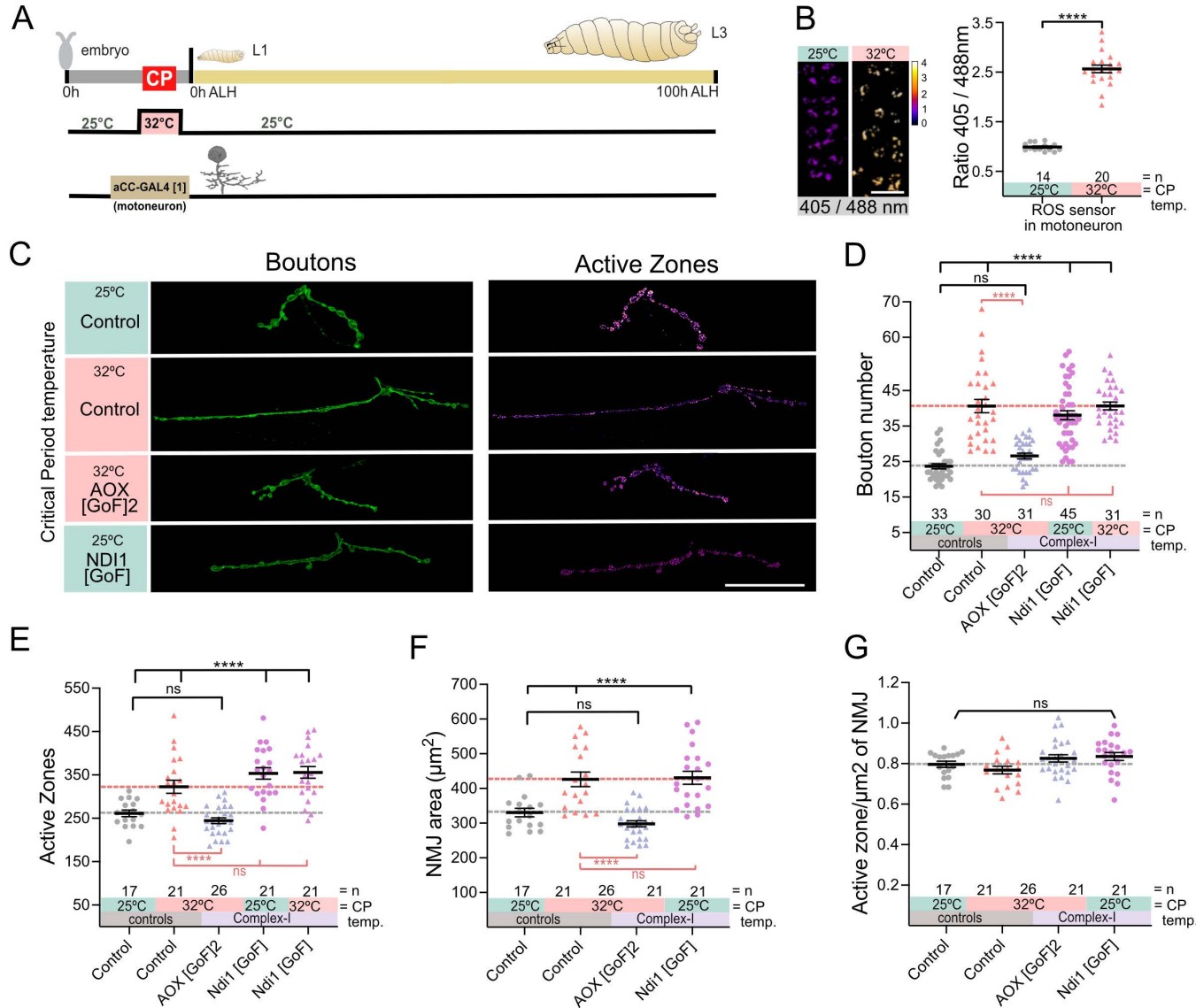

**Fig 3. ROS by RET in embryonic motoneurons instructs subsequent NMJ development. (A)** Experimental paradigm. **(B)** 32°C during the critical period increases mitochondrial ROS production in aCC motoneurons. Images generated by dividing the region of interest of 405 nm image by the same region obtained by 488 nm. Dot plots of mitochondrion-targeted ratiometric mito::roGFP2::Tsa2ΔCPΔCR ROS sensor in late stages at 25°C control temperature and 32°C. Scale bar = 10 μm. **(C)** Temperature experienced during the embryonic critical period (25°C control *vs.* 32°C heat stress) and simultaneous, transient genetic manipulation of motoneuron aCC during embryonic stages only via *aCC-GAL4*[1] (aka*RN2-O-GAL4*). "Control" indicates control genotype heterozygous for Oregon-R and *aCC-GAL4*[1]. Larvae were reared at the control temperature of 25°C until the late wandering stage, 100 h ALH. Scale bar = 20 μm. **(D)** Dot-plot quantification shows changes to aCC NMJ growth on its target muscle DA1, based on the standard measure of the number of boutons (swellings containing multiple presynaptic release sites/active zones). Data are shown with mean ± SEM, ANOVA, ****$p < 0.00001$, 'ns' indicates statistical non-significance. Black asterisks indicate comparison with the control condition of 25°C throughout, genetically unmanipulated. Red asterisks indicate comparisons with control genotype exposed to 32°C heat stress during the embryonic critical period. **(E)** Dot-plot quantification shows changes in the number of active zones at aCC NMJs quantified in (C). Data are shown with mean ± SEM, ANOVA, ****$p < 0.00001$, 'ns' indicates statistical non-significance. Black asterisks indicate comparison with the control condition of 25°C throughout, genetically unmanipulated. Red asterisks indicate comparisons with control genotype exposed to 32°C heat stress during the embryonic critical period. **(F)** Dot-plot quantification shows changes in NMJ area. Data are shown with mean ± SEM, ANOVA, ****$p < 0.00001$, 'ns' indicates statistical non-significance. Black asterisks indicate comparison with the control condition of 25°C throughout, genetically unmanipulated. Red asterisks indicate comparisons with control genotype exposed to 32°C heat stress during the embryonic critical period. **(G)** Dot-plot quantification shows no changes in active zones density (active zone/μm$^2$). Data are shown with mean ± SEM, 'ns' indicates statistical non-significance. Comparison with the control condition of 25°C throughout, genetically unmanipulated. See raw data in S3 Data.

phenotypes caused by critical period heat stress or ROS manipulations could be instigated as well as rescued through either presynaptic or postsynaptic processes. Interestingly, motoneuron manipulations during the critical period did not affect GluRII composition in muscles (S2 Fig; S7 Data).

Next, we tested the importance of developmental timing during which transient RET-generated ROS can cause such lasting change, i.e., whether mitochondrial ROS are indeed critical period-associated signals. To this end, we contrasted the effects of transient genetic manipulations of aCC motoneurons, during the late embryonic stage, including the critical period, *versus* identical manipulations after the critical period has closed (i.e., during the first larval stage). We used another, independently generated aCC expression line, "aCC-GAL4[2]" (aka *GMR96G06-GAL4*). aCC-GAL4[2] targets GAL4 to the aCC motoneuron from 2.5h prior to critical period opening and then maintains expression during larval stages (S3 Fig) [41,42]. We achieved temporal control of GAL4 activity, independent of temperature, by combining aCC-GAL4[2] with the auxin-sensitive GAL4 inhibitor from the recently published "AGES" system [43]. To induce GAL4 activity during the embryonic critical period, we fed auxin to gravid females (thus introduced into eggs and embryos), while for later GAL4 expression, postcritical period closure, we fed auxin to freshly hatched larvae for 24h. The results confirm that the timing of mitochondrial RET-generated ROS is important: only when cells generate mitochondrial ROS signals during the embryonic phase, including the critical period, do ROS cause lasting change, in keeping with this being a critical period of locomotor network development (S4 Fig; S8 Data).

## The conserved HIF-1α pathway transduces the mitochondrial ROS signal to the nucleus

Then, we considered how mitochondrial ROS, triggered by heat stress, might effect change to developmental outcomes. HIF-1α is a transcription factor that plays a central role in cellular responses to hypoxia and other stresses, including oxidative stress [44]. Using endogenous HIF-1α tagged with GFP, we observed that both exposure to 32°C or mis-expression of Ndi1 lead to an increase in cytoplasmic HIF-1α::GFP and translocation to the nucleus, both in muscles and in motoneurons (Fig 4A). We asked if HIF-1α is indeed an instructive signal during the critical period. First, we knocked down the single *Drosophila* homologue of *HIF-1α*, *sima,* selectively in DA1 muscles while exposing embryos to 32°C during the critical period. This showed that HIF-1α is necessary to cause heat stress-induced critical period plasticity, e.g., NMJ overgrowth (Fig 4A–4D; S4 Data) and a postsynaptic decrease of the GluRIIA receptor subunit (Fig 4E; S4 Data). Conversely, transient embryonic mis-expression of *HIF-1α* in DA1 muscles, at the control temperature of 25°C, is sufficient to induce these NMJ phenotypes (Fig 4). Under normoxic conditions, HIF-1α is continuously degraded by the proteasome, regulated via hydroxylation by the conserved prolyl hydroxylase, PHD (a single *Drosophila* homologue, *Hph/fatiga*). Hydroxylated HIF-1α is recognized by the E3 ligase complex substrate recognition subunit, von Hippel-Lindau (VHL), leading to HIF-1α ubiquitination and subsequent proteasomal degradation [45]. Mitochondrial ROS, including RET-generated, can inhibit PHD and thereby stabilize HIF-1α, also under normoxic conditions [23,46–49]. To consolidate the role of HIF-1α signaling, we elicited transient stabilization of HIF-1α in embryonic muscle DA1 at the control temperature of 25°C, via RNAi knockdown of HIF-1α degradation machinery components: the prolyl hydroxylase, PHD, or the VHL ubiquitin ligase complex substrate recognition subunit. Both manipulations phenocopy the characteristic critical period-induced NMJ overgrowth phenotype (Fig 4).

To further validate a working model of HIF-1α signaling as downstream of mitochondrial RET-generated ROS, we used a genetic epistasis experiment: inducing RET via mis-expression of Ndi1, while simultaneously knocking down the putative downstream acceptor, HIF-1α. We found that RET induction was no longer able to cause the expected changes in the larval NMJ when HIF-1α was knocked down at the same time (Fig 4). This supports a working model of information flow from mitochondrial RET-generated ROS to nuclear HIF-1α.

Testing this model of information flow in motoneurons, we selectively knocked down HIF-1α in motoneurons during critical period heat stress, which generates ROS via RET. This manipulation significantly suppressed both critical

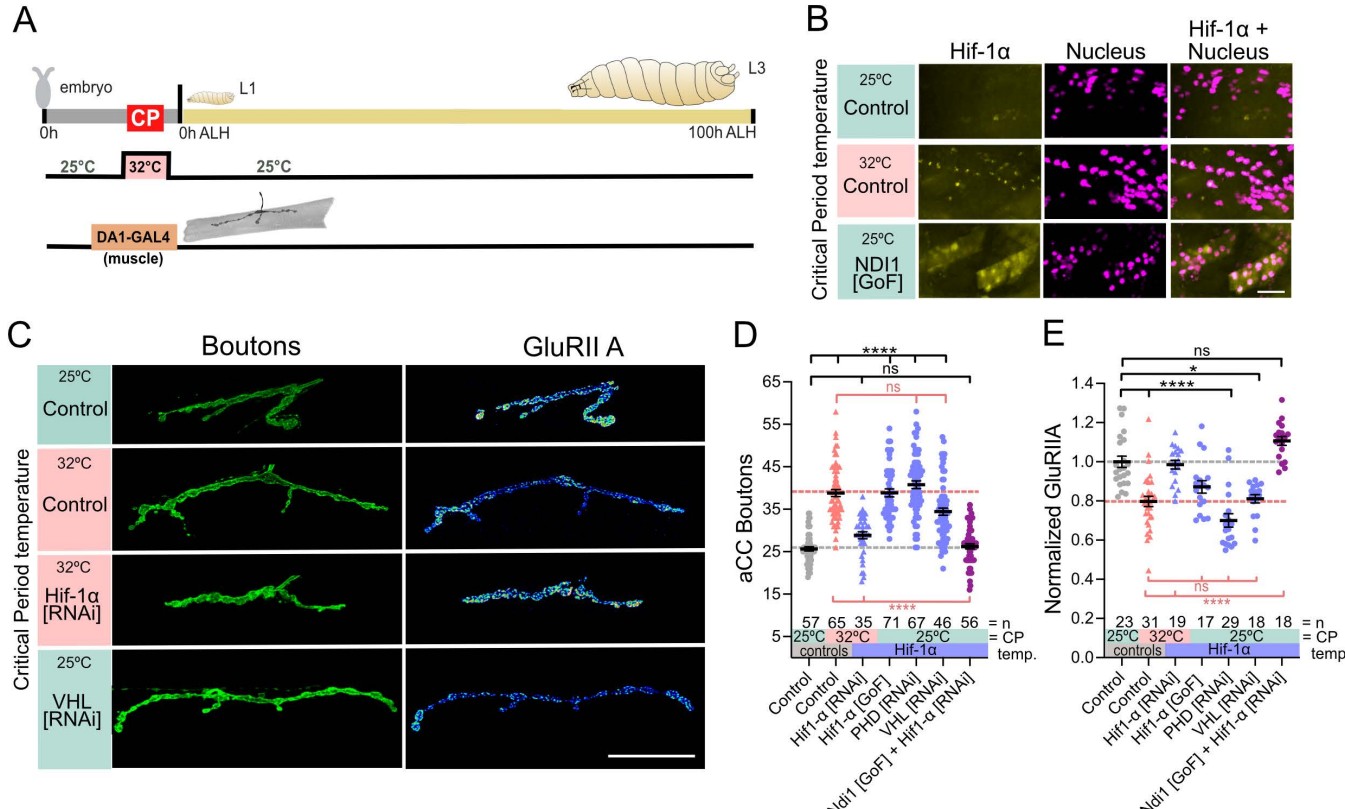

**Fig 4. HIF-1α is the signal downstream of ROS-RET. (A)** Experimental paradigm. **(B)** HIF-1α is stabilized and translocated to nucleus in muscles following 32°C or increase ROS by mitochondrial Complex-I (Ndi1 mis-expression) during the critical period. Images show GFP tagged endogenous HIF-1α and a nuclear expression of a red fluorescent protein. Scale bar = 100 μm. **(C)** Temperature experienced during the embryonic critical period (25°C control *vs.* 32°C heat stress) and simultaneous genetic manipulation of muscle DA1 during embryonic stages only. "Control" indicates control genotype heterozygous for Oregon-R and *DA1-GAL4*. Larvae were reared at the control temperature of 25°C until the late wandering stage, 100h ALH. GluRIIA subunit is displayed with a lookup table "fire" to illustrate signal intensities (warmer colors indicating greater signal intensities). Scale bar = 20 μm. **(D)** Dot-plot quantification shows changes to aCC NMJ growth on its target muscle DA1, based on the standard measure of the number of boutons (swellings containing multiple presynaptic release sites/active zones). Data are shown with mean ± SEM, ANOVA, ****$p < 0.00001$, 'ns' indicates statistical non-significance. Black asterisks indicate comparison with the control condition of 25°C throughout, genetically unmanipulated. Red asterisks indicate comparisons with control genotype exposed to 32°C heat stress during the embryonic critical period. **(E)** Dot-plot quantification shows changes in levels of the GluRIIA receptor subunit at aCC NMJs quantified in (D). Data are shown with mean ± SEM, ANOVA, *$p < 0.01$, ****$p < 0.00001$, 'ns' indicates statistical non-significance. Black asterisks indicate comparison with the control condition of 25°C throughout, genetically unmanipulated. Red asterisks indicate comparisons with control genotype exposed to 32°C heat stress during the embryonic critical period. See raw data in S4 Data.

period heat stress-induced NMJ overgrowth and the concomitant increase in active zones. Thus, in motoneurons as in muscles, HIF-1α is required for critical period plasticity (Fig 5A and 5B; S5 Data). Given that all muscle-targeted manipulations of HIF-1α gain-of-function and stabilization by inhibition of degradation produced comparable phenotypes, we proceeded with the most effective approach: co-expression of HIF-1α gain-of-function together with RNAi against VHL. This was sufficient to fully recapitulate the heat-induced NMJ phenotype at the control temperature of 25°C (Fig 5A–5D; S5 Data). We further challenged the working model by carrying out a genetic epistasis experiment, of inducing RET via mis-expression of Ndi1, while simultaneously knocking down HIF-1α. This did not have an impact at NMJ, reinforcing the conclusion that HIF-1α is the key effector downstream of mitochondrial ROS in mediating critical period plasticity.

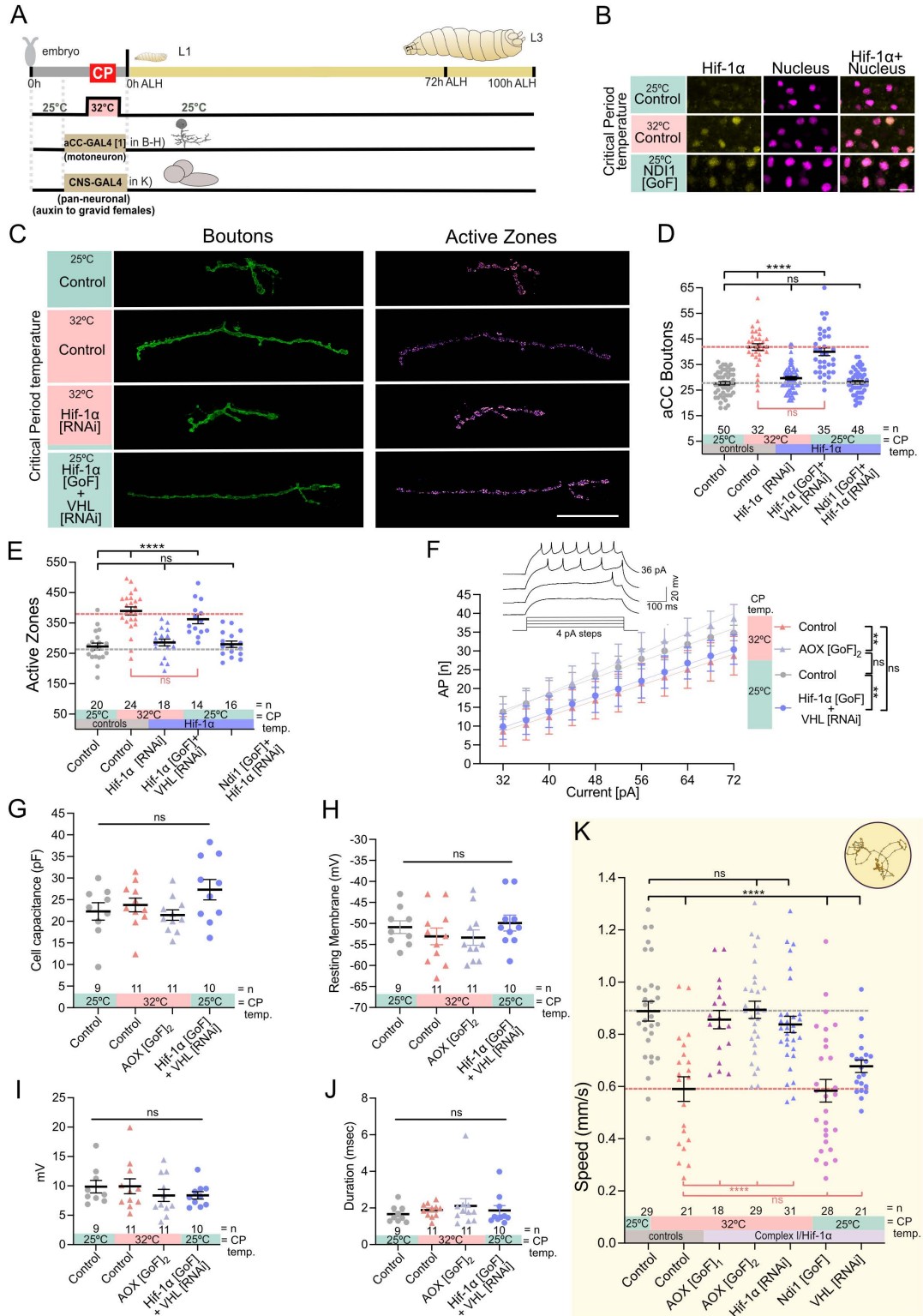

**Fig 5. ROS by RET and HIF-1α signaling during the embryonic critical period is necessary and sufficient to reduce motoneuron excitability and crawling behavior. (A)** Experimental paradigm. **(B)** HIF-1α is stabilized and translocated to nucleus in motoneurons following 32°C or increase ROS by mitochondrial Complex-I (Ndi1 mis-expression) during the critical period. Images show GFP tagged endogenous HIF-1α and a nuclear

expression of a red fluorescent protein. Scale bar = 10 μm. **(C)** Temperature experienced during the embryonic critical period (25°C control *vs.* 32°C heat stress) and simultaneous genetic manipulation of motoneuron aCC during embryonic stages only. "Control" indicates control genotype heterozygous for Oregon-R and *aCC-GAL4[1]*. Larvae were reared at the control temperature of 25°C until the late wandering stage, 100 h ALH. Scale bar = 20 μm. **(D)** Dot-plot quantification shows changes to aCC NMJ growth on its target muscle DA1, based on the standard measure of the number of boutons (swellings containing multiple presynaptic release sites/active zones). Data are shown with mean ± SEM, ANOVA, ****$p < 0.00001$, 'ns' indicates statistical non-significance. Black asterisks indicate comparison with the control condition of 25°C throughout, genetically unmanipulated. Red asterisks indicate comparisons with control genotype exposed to 32°C heat stress during the embryonic critical period. **(E)** Dot-plot quantification shows changes in levels of the active zones in aCC motoneurons quantified in (D). Data are shown with mean ± SEM, ANOVA, *$p < 0.01$, ****$p < 0.00001$, 'ns' indicates statistical non-significance. Black asterisks indicate comparison with the control condition of 25°C throughout, genetically unmanipulated. Red asterisks indicate comparisons with control genotype exposed to 32°C heat stress during the embryonic critical period. **(F)** aCC motoneuron excitability at the late third instar larval stage (i.e., action potential firing frequency triggered by current injected). Temperature experienced during the embryonic critical period (25°C control *vs.* 32°C heat stress) and simultaneous transient genetic manipulation of aCC motoneuron during embryonic stages, via *aCC-GAL4[1]* (*RN2-O-GAL4*). "Control" indicates control genotype heterozygous for Oregon-R and aCC-GAL4[1]. Larvae were reared at the control temperature of 25°C until the late wandering stage, c. 100 h ALH. Control at 25°C *vs.* 32°C is significant at $p = 0.0005$; Control at 25°C *vs.* HIF-1α[GoF] is significant at $p = 0.0015$; Control at 25°C *vs.* AOX[GoF]$_2$ at 32°C is not significant at $p = 0.28$. **(G)** Cell capacitance measurements as indicators of cell size show no significant differences between aCC motoneurons from specimens in (F). **(H)** Resting membrane potential, as an indicator for cell integrity, shows no significant differences between aCC motoneurons from specimens in (F). **(I)** Quantification of the first action potential amplitude (in mV) recorded from aCC motoneurons from specimens in (F). **(J)** Duration of first action potential fired (measured at 50% relative amplitude) shows no significant differences between aCC motoneurons from specimens in (F). **(K)** Crawling speed of third instar larvae (72 h ALH). Temperature experienced during the embryonic critical period (25°C control *vs.* 32°C heat stress) and simultaneous genetic manipulation of CNS neurons during embryonic stages only. "Control" indicates control genotype heterozygous for Oregon-R and *CNS-GAL4; Auxin-GAL80*. Larvae were reared at the control temperature of 25°C. All animals, including "Control" are from gravid females fed with auxin. Each data point represents the crawling speed from an individual uninterrupted continuous forward crawl, *n* = specimen replicate number, up to three crawls assayed for each larva. Mean ± SEM, ANOVA, ns = not significant, ****$p < 0.00001$. See raw data in S5 Data.

## Neuronal excitability and crawling speed are specified during the critical period by the mitochondrial ROS to HIF-1α signaling axis

Having thus far focused on the NMJ as a model synapse, we asked whether the structural changes that result from critical period plasticity have correlates in neuronal function and behavior [50]. To this end, we performed whole-cell patch clamp recordings from the aCC motoneuron at the late larval stage. These showed that embryonic heat stress of 32°C caused a lasting reduction in excitability (i.e., reduced capability to fire action potentials). Though cell size (assessed by membrane capacitance) and resting membrane potential were not affected. We also analyzed the first action potential fired by the lowest current injection: both the amplitude and the duration at half-maximum amplitude remained unchanged (Fig 5E–5H; S5 Data). Thus, we are not yet able to propose a mechanism that underlies the observed reduction in excitability. This is further compounded by the fact that the action potential initiation zone is very far removed from the site of recordings [51].

At a network level, exposure to 32°C during the critical period results in reduced crawling speed at the late larval stage (Fig 5K; S5 Data). We find that mitochondrial RET-ROS and HIF-1α signaling are required and sufficient for this critical period-regulated change in locomotor network output. Transient induction of these signals, in all embryonic neurons, at the control temperature, phenocopies the reduction in crawling speed caused by critical period heat stress (Fig 5K; S5 Data). However, RET-ROS and HIF-1α signaling only have this lasting effect when induced during the embryonic phase, including the critical period window. In contrast, manipulations outside this plastic period, i.e., post critical period closing, are without effect (S5 Fig; S9 Data). Thus, the changes we observed following critical period heat stress, both centrally and at the NMJ, are sufficient to alter locomotor behavior.

To understand the underlying reason, we considered two scenarios that could explain reduced locomotor speed. First, embryonic exposure to 32°C may lead to larvae that are simply unable to crawl as fast as control animals. To test this, we acutely increased the temperature during the crawling assay and found that larvae responded with an increase in speed regardless of whether or not they had been exposed to heat stress as embryos. Moreover, under such conditions, critical period-manipulated animals could match the crawling speed of controls (S6A Fig; S10 Data). As a second hypothesis, embryonic heat stress during the critical period might 'set' a different locomotor network homeostatic setpoint. To test

this, we maintained critical period-manipulated animals at 32°C for several hours. This results in animals reverting to their "default velocity", including critical period-manipulated larvae, which reverted to the original reduced level, suggesting that the system is still capable of homeostasis: responding to change and returning to a pre-determined "default velocity" (S6B Fig; S10 Data). These observations suggest, that following critical period heat stress, the system is not simply damaged, but remains responsive to environmental temperature challenges, as in control animals. We speculate that following a critical period manipulation the locomotor network "default output" is positioned at a different set point or range.

We conclude that during the embryonic critical period, mitochondrial RET-generated ROS are instructive signals that are transmitted to the nucleus via the conserved transcriptional regulator, HIF-1α. Only when activated during the embryonic critical period, does this signaling pathway alter the subsequent development of the nervous system, leading to significant and permanent change in synaptic terminal growth, receptor composition, neuronal excitable properties, and network output, which manifests as changes in larval crawling behavior (Fig 6).

## Discussion

Working with a simplified experimental system, we have identified a metabolic signal, mitochondrial RET-generated ROS, as a primary critical period plasticity signal, that is transduced to the nucleus via the conserved HIF-1α signaling pathway. Moreover, we demonstrated that critical period-plasticity responses are fundamentally cell intrinsic, shown by virtue of targeting genetic manipulations, transiently, to single cells in vivo. This provides a first mechanistic explanation for how a perturbation experienced during a critical period effects a change in subsequent development, leading to altered neuronal properties and animal behavior.

### Environmental cues are integrated during critical periods and direct nervous system development

It has long been known that early life experiences are formative, due to their potential to impart lasting influence on nervous system function [52,53]. Though critical periods have been most intensively studied in the context of activity-regulated sensory processing and cortical networks, it is possible that critical periods represent a fundamental process that is common to the development of nervous systems. For example, similar developmental windows of heightened

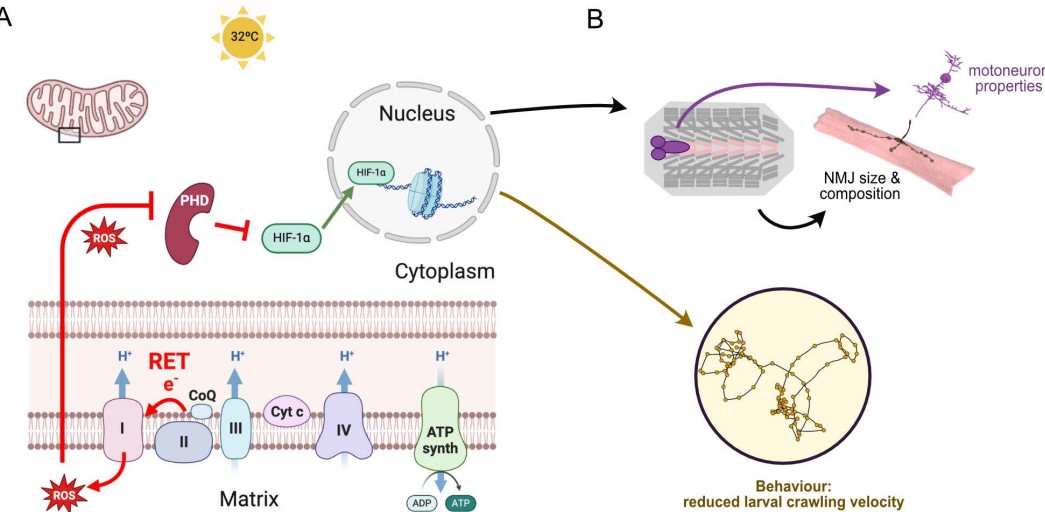

**Fig 6. Working model. (A)** Heat stress during the critical period induces mitochondrial ROS, which leads to HIF-1α stabilization and its accumulation in the nucleus, where HIF-1α generates long-lasting changes that impact on **(B)** NMJ development: larger presynaptic terminals and less GluRIIA, as well as reductions in motoneuron excitability and, behaviorally, crawling speed. Figure A created using BioRender (https://www.biorender.com/).

vulnerability to disturbances have also been identified in mammalian locomotor systems [54], as well as in locomotor, visual, and olfactory systems of zebrafish [55–59] and several insect species (ants, bees, and *Drosophila* [60–64]). In fast-developing organisms, opening of a critical period correlates with the phase when network activity transitions from spontaneous and un-patterned to patterned activity, and in the case of the locomotor network of the *Drosophila* embryo lasts only 2 h [11,13,14,65–67]. In mammalian systems, critical periods are much more drawn out, lasting weeks in mouse and up to several years in human [7]. Notwithstanding such differences, the neuronal activity-regulated processes of critical periods appear remarkably similar across species; from GABAergic signaling during critical periods of visual system and cortex development [59], to the importance of networks attaining an appropriate excitation:inhibition balance, also recently shown for the developing *Drosophila* locomotor network [12,13].

Here, we took advantage of the well-defined critical period of the *Drosophila* locomotor network [11,12]. As an ecologically relevant stimulus, we focused on temperature, specifically a 32°C heat stress, which we have previously shown phenocopies pharmacological or genetic activity manipulations, leading to similar lasting changes in subsequent neuronal development [50]. Using this experimental paradigm allowed us to identify a basic instructive signal that underlies critical period plasticity, namely mitochondrial ROS generated at Complex-I via reverse flow of electrons within the ETC. In mammals, mitochondrial RET-generated ROS has been associated with aging and expression of protective responses [19,68]. Whether this same signal is also responsible for the responses to developmental heat stress reported in ants [69], bees [62,70,71] or other parts and stages of the developing *Drosophila* nervous system [15,60], remains to be tested. Though we employed a stress signal in this study, to elicit a robust critical period plasticity response, temperature operates as a general environmental cue during nervous system development. Complementing this study, a cooler, non-stress temperature experienced during a pupal critical period of *Drosophila* nervous system development, has been reported to also lead to clear cellular effects. In that case, causing reduced filopodial dynamics and a consequent increase in synapse formation with concomitant changes in network connectivity and behavioral output [60,64].

## Critical period effects at the level of single cells

One of the strengths of the experimental system we use is that it allows studying critical periods with single-cell resolution in vivo. Here, we demonstrated that critical period plasticity is primarily cell-autonomous. This simplifies establishing clear cause–effect relationships, which is more difficult with network-wide manipulations that are commonly used. Through cell-selective genetic manipulations of a single motoneuron or its target muscle, we phenotypically rescued individual cells from systemic heat stress manipulations. Thus, we demonstrated the cell-autonomous necessity of RET-generated mitochondrial ROS and HIF-1α signaling for heat stress-induced critical period plasticity. Conversely, we also showed cell-selective sufficiency of RET-ROS and HIF-1α signaling, which can phenocopy changes in NMJ development; though only when induced during the transient embryonic critical period, but not after.

We observed that muscle manipulations during late embryogenesis alter the composition of the postsynaptic receptors within the muscle. They also affect the development of the NMJ (increase in size and number of presynaptic release sites), which arises from an interplay between presynaptic motor axon terminal and postsynaptic muscle [72]. In contrast, motoneuron-targeted critical period manipulations do not affect the postsynaptic muscle GluRII composition. During embryogenesis, the body wall muscles become electrically active 2–3 h before the motoneuron is able to conduct action potentials and receive excitatory synaptic input [65,73,74]. This sequence of muscles maturing before presynaptic motoneurons would be compatible with muscles undergoing their critical period before motoneurons, and compatible with our observation of muscles having "decided" their GluRII composition before the motoneuron can influence other aspects of NMJ formation. From a perspective of the developing nervous system, it is well known that different regions undergo their respective critical periods in sequence [7]. This may be mirrored at the cellular level within a given region or network: different neuronal types might be integrated into circuits sequentially, for example, based on birth order or maturation speed, facilitating more reproducible and robust network assembly. In support of such a view, a recent study centered on

developing mouse neocortex demonstrated the sequential maturation of two principal classes of inhibitory interneurons, and how this process directs network maturation [75].

## Is the critical period a developmental window during which network setpoints are established?

Neuronal networks dynamically maintain stability and appropriate function around a given set point (or range), through homeostatic plasticity mechanisms [76,77]. How and when a set point is specified remains unknown. Previous evidence suggests that perturbations of activity during a critical period can alter the homeostatic set point, and thus lead to long-lasting changes in network function, including instability and epilepsy-like activity [11,78,79]. Thus, the focus has been on neuronal activity and its coordination as important processes in setting the set point [11,12,65–67,80,81]. A challenge has been to relate the consequences of perturbing network activity, during a critical period, to specific changes in functional or behavioral outcomes. By working with a locomotor network, we found that heat stress at the cellular level, as a critical period perturbation, leads to reduced excitability of motoneurons. At the network level, this correlates with reduced crawling speed, which is seen as directly reflecting a change in output of the locomotor network. Importantly, changes in crawling speed only occurred following manipulations that spanned the embryonic critical period, and animals maintained the ability to increase their crawling speed. Moreover, network homeostasis appears to remain intact, as critical period-manipulated "slow" animals respond normally to acute upshifts in environmental temperature, but return to their slower "default" speed following chronic changes [30]. These observations are compatible with the possibility that the set point for locomotor network output is established during the critical period. Because this network is so well characterized and experimentally tractable, it now offers the opportunity to study the cellular mechanisms through which this is achieved.

## Conserved mitochondrial-nuclear signaling directs critical period plasticity

Cells integrate internal (e.g., network activity) and external environmental cues (e.g., temperature) during a critical period [12]. The identification of an instructive signal that is intimately linked to mitochondrial metabolism significantly adds to, and complements, a large body of work that has focused on activity-regulated processes [1,2]. How a neuronal activity-regulated signal and mitochondrial metabolism interlink, especially during critical periods of development, will be important to understand. We had previously shown that mitochondrial and NADPH oxidase-generated ROS are required for activity-regulated neuronal plasticity outside the critical period [30,31,82]. Here, we identified a specific source of mitochondrial ROS, namely those generated at Complex-I via RET, which occurs when the coenzyme Q pool is reduced while the proton motif force is elevated [83]. A mitochondrial enzyme that is linked to the redox state of the coenzyme Q pool, dihydroorotate dehydrogenase (DHODH), has been identified as an important regulator of neuronal action potential firing rate, i.e., a determinant of the neuronal homeostatic setpoint [84,85]. Together with our findings here, we speculate that mitochondrial metabolism might be central to critical period plasticity, potentially encoding neuronal homeostatic setpoints.

How might a transient perturbation during a critical period lead to lasting change? HIF-1α, a transcriptional regulator involved in the cellular response to stress, is typically degraded under normal oxygen concentrations. The model we propose is that mitochondrial ROS, including those generated by RET, transiently inhibit the degradation of HIF-1α, thus allowing HIF-1α to accumulate and translocate into the nucleus [23,46–49]. HIF-1α signaling is a known regulator of changes in cellular metabolism, for example, promoting aerobic glycolysis in a range of cancers [86–88], but also during normal development, when rapid tissue growth is required [26,89,90]. HIF-1α achieves this through a combination of direct regulation of gene expression, notably of metabolic enzymes, and epigenetic modifications, partly executed through histone acetyl transferases that act as co-factors [91–94]. In some forms of cancer, this is thought to create positive feedback loops, from elevated mitochondrial metabolism, to increased ROS production maintaining disinhibition of HIF-1α, which reinforces metabolic drive. It is therefore conceivable that transient HIF-1α signaling could initiate such metabolic-epigenetic feedback loops that maintain a change in metabolic state, and thus also of neuronal growth and functional properties.

## Materials and Methods

### *Drosophila* rearing and stocks

*Drosophila* stocks were kept on standard corn meal medium, at 25°C. For all experiments where crosses were necessary, stocks were selected and combined as a 1:2 ratio of GAL4-line males to UAS-line virgin females. The following *Drosophila melanogaster* strains were obtained from the Bloomington Stock Center: Oregon-R-P2 (BDSC_2376), *Mef2-GAL4* (BDSC_27390), *VNC-GAL4* (BDSC_40701), *aCC-GAL4[2]* (BDSC_39171), *UAS-myr-GFP* (32196), *Hif-1α::GFP* (aka sima-GFP) (BDSC_59821), *UAS-RedStinger* (BDSC_8546), *UAS-Hif-1α[RNAi]* (BDSC_26207), *UAS-VHL[RNAi]* (BDSC_50727), and *UAS-PHD[RNAi]* (BDSC_34717). The following strains were kindly provided as gifts: *aCC-GAL4[1]* (aka *RN2-O-GAL4* or *eve-GAL4.RN2-O*) [95] and *DA1-GAL4* (aka *eve-GAL4.eme*) [96] from Miki Fujioka; AtTIR1-T2A-AID-GAL80-AID inserted into attP-3B (from Tony Southall) [43], *UAS-mito-catalase* (from Alex Whitworth) [97], *UAS-Prx3*, *UAS-ND75[RNAi]*, *UAS-AOX[GoF]₁* and *UAS-AOX [GoF]₂*, *UAS-Ndi1[GoF]* (all from Alberto Sanz), and *UAS-Hif-1α[GoF]* (aka *UAS-sima*; from Joe Bateman). We generated the *UAS-mito::roGFP2::Tsa2 ΔCPΔCR* ROS sensor (see below). See Table 1 for more details of the strains used.

### Critical period manipulations

**Temperature manipulations.** Embryos were harvested from 0 to 6 h after egg lay at 25°C. Embryos were then shifted to their respective temperatures in incubators (25 or 32°C) for the remainder of embryogenesis. After all animals hatched, all were moved back to 25°C until assayed in late larval stages.

**Auxin feeding.** Auxin (GK2088 1-Naphthaleneacetic acid potassium salt; Glentham) solution was mixed with melted molasses fly medium to achieve 5 mM auxin concentration and poured into petri dish plates. For embryonic exposure to auxin, adult flies in mixed-sex laying pots were fed on this food for 48 h prior to egg collection. For larval exposure, freshly hatched larvae were fed on this food for 24 h.

### Dissections and immunocytochemistry

Flies were allowed to lay eggs on apple-juice agar-based medium overnight at 25°C. Larvae were then reared at 25°C on yeast paste, while avoiding over-crowding. Precise staging of the late wandering third instar stage was achieved by: a) checking that a proportion of animals from the same time-restricted egg lay had initiated pupariation; b) larvae had reached a certain size, and c) showed gut-clearance of food (yeast paste supplemented with Bromophenol Blue Sodium Salt (Sigma-Aldrich)). Larvae (of either sex) were dissected in Sorensen's saline, fixed for 10 min at room temperature in Bouins fixative (Sigma-Aldrich), as previously detailed [31]. Wash solution was Sorensen's saline containing 0.3% Triton X-100 (Sigma-Aldrich) and 0.25% BSA (Sigma-Aldrich). Primary antibodies, incubated overnight at 10°C, were: Goat-anti-HRP Alexa Fluor 488 (1:1000; Jackson ImmunoResearch Cat. No. 123-545-021), Rabbit-anti-GluRIIB (1:1000; gift from Mihaela Serpe [98]), Mouse anti-GluRIIA (1:100; Developmental Studies Hybridoma Bank, Cat. No. 8B4D2 (MH2B)); Mouse anti-BRP (1:100; Developmental Studies Hybridoma Bank, Cat. No. nc82); secondary antibodies, 2 h at room temperature: Donkey anti-Mouse StarRed (1:1000; Abberior, Cat. No. STRED-1001); and goat anti-Rabbit Atto594 (1:1000; Sigma-Aldrich Cat. No. 77671-1ML-F). Specimens were cleared in 80% glycerol, overnight at 4°C, then mounted in Mowiol.

### Image acquisition and analysis

Specimens were imaged using an Olympus FV3000 point-scanning confocal, and a 60x/1.3N. A silicone oil immersion objective lens and FV31S-SW software. Confocal images were processed using ImageJ (to quantify GluRIIA and -B intensity) and Affinity Photo (Serif Ltd) to prepare figures. Bouton number of the aCC NMJ on muscle DA1 from segments A3-A5 was determined by counting every distinct spherical varicosity.

**Table 1. *Drosophila* strains.**

| Drosophila strain genotype | Additional information | FlyBase identifiers | Source |
|---|---|---|---|
| *Oregon-R-P2* | Wild Type | FBst0002376 | Bloomington Drosophila Stock Center; RRID: BDSC_2376 |
| *eve-GAL4[eme-GAL4] (aka SS2-C-GAL4; referred to as "DA1-GAL4")* | GAL4 expression targeted to muscle DA1 during embryonic stages only | FBtp0016392 | Gift from Miki Fujioka; described in Han and Colleagues [96] |
| *Mef2-GAL4* | GAL4 expression targeted to all muscles | FBst0027390 | Bloomington Drosophila Stock Center; RRID: BDSC_27390 |
| *RN2-O-GAL4 (aka eve-GAL4.RN2}O; referred to as "aCC-GAL4 [1]")* | GAL4 expression targeted to motorneurons aCC and RP2 during embryonic stages only | FBti0038624 | Gift from Miki Fujioka; generated by Fujioka and Colleagues [95] |
| *GMR94G06-GAL4 (referred to as "aCC-GAL4 [2]")* | GAL4 expression targeted to motorneuron aCC | FBti0139793 | Bloomington Drosophila Stock Center; RRID: BDSC_40701 |
| *GMR57C10-GAL4, AtTIR1-T2A-AID-Gal80-AID (aka n-synaptobrevin-GAL4; referred to as CNS-GAL4)* | GAL4 expression targeted to neurons | FBti0137043 | Bloomington Drosophila Stock Center; RRID: BDSC_39171 |
| *AtTIR1-T2A-AID-GAL80-AID (referred to as Auxin-GAL80)* | Ubiquitously expresses Gal80 sensitive to auxin | FBti0216871 | Bloomington Drosophila Stock Center; RRID: BDSC_92470; gift from Tony Southall |
| *10xUAS-IVS-mito::roGFP2::Tsa2 ΔCPΔCR (referred to as UAS-mito::roGFP2::Tsa2 ΔCPΔCR)* | Mitochondrial targeted ROS sensor | ------------ | This paper: see methods |
| *UAS-mito-catalase* | Mitochondrial-targeted catalase under UAS control | FBtp0057181 | Gift from Alex Whitworth; generated by Radyuk and Colleagues [97] |
| *UAS-Prx3* | Peroxiredoxin 3 under UAS control | FBtp0057179 | Gift from Alberto Sanz |
| *UAS-ND75 [RNAi]* | Expresses dsRNA for RNAi of *ND75* (FBgn0017566) under UAS control | FBst0472606 | Vienna Drosophila Resource Center #100733/KK |
| *UAS-AOX [GoF]1* | Expresses AOX from *C. intestinalis* under UAS control (allele F6) | FBti0147237 | Gift from Alberto Sanz |
| *UAS-AOX [GoF]2* | Expresses AOX from *C. intestinalis* under UAS control (allele F24) | FBti0147239 | Gift from Alberto Sanz |
| *UAS-Ndi1 [GoF]* | Expresses Ndi1 from *S.cerevisae* under UAS control (allele A46 and B20) | FBti0182574 and FBti0182575 | Gift from Alberto Sanz |
| *UAS-myrGFP* | Expresses GFP tagged with a myristoylation site under the control of UAS sequences | FBst0032196 | Bloomington Drosophila Stock Center; RRID: BDSC_32196 |
| *Hif-1-GFP* | GFP tagged at native locus of *Hif-1* | FBti0178346 | Bloomington Drosophila Stock Center; RRID: BDSC_59821 |
| *UAS-nRFP* | Expresses a variant DsRed protein with a nuclear localization signal under the control of UAS | FBti0040829 | Bloomington Drosophila Stock Center; RRID: BDSC_8546 |
| *UAS-Hif-1 [RNAi]* | Expresses dsRNA for RNAi of *sima* (FBgn0266411) under UAS control | FBti0115271 | Bloomington Drosophila Stock Center; RRID: BDSC_26207 |
| *UAS-Hif-1 [GoF] (aka UAS-sima)* | Expresses Hif-1/*sima* (FBgn0266411) under UAS control | FBti0076469 | Bloomington Drosophila Stock Center; RRID: BDSC_9582 |
| *UAS-VHL [RNAi]* | Expresses dsRNA for RNAi of *VHL* (FBgn0041174) under UAS control | FBti0158781 | Bloomington Drosophila Stock Center; RRID: BDSC_50727 |
| *UAS-PHD [RNAi]* | Expresses dsRNA for RNAi of *PHD* (FBgn0264785) under UAS control | FBti0140885 | Bloomington Drosophila Stock Center; RRID: BDSC_34717 |

To quantify GluRIIA and -B fluorescence intensity, a threshold image for each channel was used to generate an outline selection, which included only the postsynaptic glutamate receptor clusters. This outline was then applied to the original image to define the regions of interest for analysis. Fluorescence intensities for GluRIIA and GluRIIB channels were measured within these outlines. Data were exported into Excel and the intensity values normalized to the negative control condition (25°C through all embryogenesis).

## Generation of a ratiometric ROS reporter

The pJFRC12-10XUAS-IVS-mito-roGFP2::Tsa2ΔCPΔCR (for simplicity referred to as UAS-mito-roGFP2::Tsa2ΔCPΔCR) transgene was generated by Klenow assembly cloning (tinyurl.com/4r99uv8m). Briefly, we used pJFRC12-10XUAS-IVS-myr-GFP plasmid DNA, removed the coding sequence for *myr::GFP* using XhoI and XbaI, and replaced it with the mito-chondrial targeting sequence from human COX8, derived from plasmid pMitoTimer [99], a gift from Zhen Yan (Addgene plasmid # 52659; http://n2t.net/addgene:52659; RRID: Addgene_52659); followed in frame by the coding sequence for roGFP2::Tsa2ΔCPΔCR from plasmid p415 TEF roGFP2-Tsa2ΔCPΔCR [33], a gift from Tobias Dick (Addgene plasmid # 83239; http://n2t.net/addgene:83239; RRID: Addgene_83239). Transgenics were generated by microinjection via phiC31 integrase-mediated recombination [100] into landing site VK00037 [cytogenetic 22A3] by the Injection Service, Department of Genetics, University of Cambridge.

## Fluorescence measurement of the ratiometric ROS reporter

Mef2-GAL4 was used for targeting expression of UAS-mito-roGFP2::Tsa2ΔCPΔCR in all muscles. Embryos from 25°C critical period (control) or 32°C critical period manipulations were dechorionated using bleach (3 min at room temperature). Late-stage embryos were selected, transferred onto double-sided sticky tape on glass slide, and covered with PBS, then imaged immediately with a Leica Stellaris point-scanning confocal, using a 63×/0.9NA (Leica) dipping objective lens. The ROS reporter was excited sequentially at 405 and 488 nm with emission detected at 500–600 nm. 16-bit images were acquired using Leica LAS X software and processed with ImageJ. Z-stack images were maximally projected. To remove fringing artifacts around bouton edges, 488 nm images were thresholded using the "Otsu" algorithm with values below threshold set to 'not a number', the mean value obtained from the same region of interest was taken for the 405 nm image and divided by the 488 nm image.

aCC-GAL4[1] was used to drive the expression of UAS-mito-roGFP2::Tsa2ΔCPΔCR in all aCC and RP2 motoneurons. Ventral nerve chords of freshly hatched (up to 2 h) larvae from 25°C critical period (control) or 32°C critical period manipulations were dissected in PBS-NEM (137 mM NaCl, 2.7 mM KCl, 10 mM Na2HPO4, 1.8 mM KH2PO4, 20 mM N-ethyl-maleimide (NEM), pH 7.4). Ventral nerve cord preparations were incubated for 5 min in PBS-NEM, then fixed for 8 min in 4% formaldehyde (in PBS-NEM). Specimens were washed three times in PBS-NEM and then equilibrated in 70% glycerol. Specimens were mounted in glycerol and imaged the same day on a Leica Stellaris point-scanning confocal, using a 40×/1.25NA (Leica) glycerol immersion objective lens. Analysis was carried out as per above.

## aCC-GAL4[2] expression imaging

Embryos containing one copy of the aCC-GAL4[2] transgene and UAS-myr-GFP transgene reporter were dechorionated using bleach (3 min at room temperature). Specific development time points embryos were selected and fixed with 4% paraformaldehyde (Agar Scientific) in saline for 15 min, at room temperature. Following the standard procedures of washes in PBS containing 0.3% Triton X-100 and 0.25% (w/v) bovine serum albumin (Sigma-Aldrich), GAL4-directed expression of membrane-targeted GFP was visualized by FluoTag-X4 single-domain anti-GFP antibody conjugated with Atto488 fluorophore (NanoTag Biotechnologies GmbH, Cat no: N0304-At488-L) and the longitudinal Fasciclin II-positive axon tracts via Mouse anti-Fasciclin II (1:10; Developmental Studies Hybridoma Bank, Cat. No. 1D4 anti-Fasciclin II) with

the secondary Donkey anti-Mouse StarRed (1:1000; Abberior, cat. no. STRED-1001). Embryos were transferred onto double-sided sticky tape on glass slide for imaging.

For image GAL4 expression in larva, two time points were selected: 2 h after hatch and 72 h after hatch. Each larva was dissected in extracellular saline to isolate the CNS, using a hypodermic syringe needle (30 gauge; BD Microlance) as a scalpel. The CNS was then transferred onto poly-l-lysine-coated (Sigma-Aldrich) cover glass. Nerve cords were fixed and stained as per embryos, cleared in 70% glycerol, then mounted in Mowiol and sandwiched under a second cover glass, with thin aluminum foil strips used as spacers. Images were acquired using a Leica Stellaris point-scanning confocal with a Leica LAS X software and processed with ImageJ. Z-stack.

### HIF-1α detection

We used a MiMIC-based RMCE line in which endogenous Hif-1α (aka Sima) was fused in-frame with GFP (see Table 1). To test the effect of Ndi1 [GoF], we crossed the Hif-1α-GFP stock with either DA1-GAL4 or aCC-GAL4[1] recombined with a nuclear-targeted red fluorescent protein reporter to mark manipulated cells. To detect expression in muscles, embryos from 25°C critical period (control) or 32°C critical period or Ndi1 [GoF] manipulations were dechorionated and mounted in saline as per above shortly after critical period closure (tracheal filling), then imaged with a CSU-22 spinning disc confocal, using an Olympus 63×/0.8NA dipping objective lens. For motoneuron analysis of Hif-1α expression, ventral nerve chords of freshly hatched larvae from 25°C critical period (control), 32°C critical period or Ndi1 [GoF] manipulations were dissected in Sorensen's saline and transferred to a cover glass coated with poly-L-lysine (Sigma-Aldrich). Ventral nerve cord preparations were then fixed for 8 min in 4% formaldehyde (in PBS). Specimens were washed three times in PBS and then equilibrated in 70% glycerol and imaged the same day on a Leica Stellaris point-scanning confocal, using a 40×/NA1.25 (Leica) glycerol immersion objective lens.

### Electrophysiology

Third instar (L3) larvae (of either sex) were dissected in a dish containing standard saline (135 mM NaCl (Fisher Scientific), 5 mM KCl (Fisher Scientific), 4 mM $MgCl_2 \cdot 6H_2O$ (Sigma-Aldrich), 2 mM $CaCl_2 \cdot 2H_2O$ (Fisher Scientific), 5 mM TES (Sigma-Aldrich), 36 mM sucrose (Fisher Scientific), pH 7.15) to remove the ventral nerve cord and brain lobes (CNS). The isolated CNS was then transferred to a droplet of external saline containing 200 µM mecamylamine (Sigma-Aldrich) to block postsynaptic nACh receptors in order to synaptically isolate motor neurons. CNSs were laid flat (dorsal side up) and glued (GLUture Topical Tissue Adhesive; World Precision Instruments USA) to a Sylgard-coated cover slip (1–2 mm depth of cured SYLGARD Elastomer (Dow-Corning USA) on a 22 × 22 mm square coverslip. Preparations were placed on a glass slide under a microscope (Olympus BX51-WI), viewed using a 60× water-dipping lens. To access nerve cell somata, 1% protease (Streptomyces griseus, Type XIV, Sigma-Aldrich, in external saline) contained within a wide-bore glass pipette (GC100TF-10; Harvard Apparatus UK, ~10 µm opening) was applied to abdominal segments, roughly between A5-A2 [101] to remove overlaying glia. Motoneurons were identified by anatomical position and relative cell size, with aCC being positioned close to the midline and containing both an ipsilateral and contralateral projection. Whole cell patch clamp recordings were made using borosilicate glass pipettes (GC100F-10, Harvard Apparatus) that were fire polished to resistances of 10–15 MΩ when filled with intracellular saline (140 mM potassium-D-gluconate (Sigma-Aldrich), 2m M $MgCl_2 \cdot 6H_2O$ (Sigma-Aldrich), 2 mM EGTA (Sigma-Aldrich), 5 mM KCl (Fisher Scientific), and 20 mM HEPES (Sigma-Aldrich), (pH 7.4). Input resistance was measured in the 'whole cell' configuration, and only cells that had an input resistance ≥0.5 GΩ were used for experiments. Cell capacitance and break-in resting membrane potential were measured for each cell recorded; only cells with a membrane potential upon break in of <−40 mV were analyzed. Data for current step recordings were captured using a Multiclamp 700B amplifier (Molecular Devices) controlled by pCLAMP (version 10.7.0.3), via an analogue-to-digital converter (Digidata 1440A, Molecular Devices). Recordings were sampled at 20 kHz and filtered online at 10 kHz. Once patched, neurons were brought to a membrane potential of −60 mV using current

injection. Each recording consisted of 20 × 4 pA (500 ms) current steps, including an initial negative step, giving a range of −4 to +72 pA. The number of action potentials was counted at each step and plotted against injected current. Cell capacitance and membrane potential were measured between conditions to ensure that any observed differences in excitability were not due to differences in either cell size or resting state.

### Larval crawling analysis

At the mid-third instar stage, 72 h after larval hatching (ALH), larvae were rinsed in water and placed inside a 24 cm × 24 cm crawling arena with a base of 5 mm thick 0.8% agarose gel (Bacto Agar), situated inside an incubator. Temperature was maintained at 25 ± 0.5°C, reported via a temperature probe in the agar medium. Humidity was kept constant. Larval crawling was recorded using a frustrated total internal reflection-based imaging method (FIM) in conjunction with the tracking software FIMTrack [102,103] using a Basler acA2040-180km CMOS camera with a 16 mm KOWA-IJM3sHC.SW-VIS-NIR Lens controlled by Pylon (by Basler) and Streampix (v.6) software (by NorPix). Larvae were recorded for 20 min at five frames per second.

Recordings were split into four 5-min sections with the first section not tracked and analyzed as larvae are acclimatizing to the crawling arena. The remaining three sections were used to analyze crawling speed, with each larva sampled at most once per section. Crawling speed was calculated using FIMTrack software choosing crawling tracks that showed periods of uninterrupted forward crawling devoid of pauses, turning, or collision events. The distance traveled was determined via the FIMTrack output.

### Statistical analysis

Statistical analyses were carried out using GraphPad Prism Software (Version 10.1.2). Datasets were tested for normal distribution with the Shapiro-Wilk Test. Normally distributed data were then tested with student $t$ test (for pairwise comparison). Normally distributed analysis for more than two groups was done with a one-way ANOVA and post hoc tested with a Tukey multiple comparison test. Non-normally distributed data sets of two groups were tested with Mann–Whitney $U$ Test (pairwise comparison) and datasets with more than two groups were tested with a Kruskal–Wallis ANOVA and post hoc tested with a Dunn's multiple post hoc comparison test. For whole cell electrophysiology in aCC neurons, linear regression analysis was used to compare the intercepts between conditions, using a Bonferroni correction for multiple comparisons, resulting in an adjusted significance threshold of $p < 0.0083$. This analysis was restricted to the linear portions of the input-output curves. For all datasets mean and standard error of mean (SEM) are shown. Significance levels were *$p < 0.05$; **$p < 0.01$; ***$p < 0.001$; ****$p < 0.0001$.

### Supporting information

**S1 Fig. Control of UAS-lines crossed to wild type Oregon-R. (A)** Dot-plot quantification shows changes to aCC NMJ growth on its target muscle DA1, based on the standard measure of the number of boutons (swellings containing multiple presynaptic release sites/active zones). Data are shown with mean ± SEM, ANOVA, ****$p < 0.00001$, 'ns' indicates statistical non-significance. Black asterisks indicate comparison with the wild type of condition of 25°C throughout, genetically unmanipulated. Red asterisks indicate comparisons with wild type exposed to 32°C heat stress during the embryonic critical period. "Wild type" is Oregon-R. **(B)** Dot-plot quantification shows changes in levels of the GluRIIA receptor subunit at aCC NMJs quantified in C). Data are shown with mean ± SEM, ANOVA, ****$p < 0.00001$, 'ns' indicates statistical non-significance. Black asterisks indicate comparison with the wild type condition of 25°C throughout, genetically unmanipulated. Red asterisks indicate comparisons with wild type exposed to 32°C heat stress during the embryonic critical period. "Wild type" is Oregon-R. See raw data in S6 Data.
(TIFF)

**S2 Fig. Muscles influence motoneuron at NMJ during the critical period, but motoneuron do not affect GluR composition in muscles. (A)** Heat stress experienced during the embryonic critical period (32°C vs. 25°C control) leads to decreased postsynaptic GluRIIA. Simultaneous genetic manipulation of aCC motoneuron during embryonic stages does not affect GluRIIA in muscles. "Control" indicates control genotype heterozygous for Oregon-R and *aCC-GAL4[1]*. Larvae were reared at the control temperature of 25°C until the late wandering stage, 100 h after larval hatching (ALH). Data are shown with mean ± SEM, ANOVA, **$p < 0.001$, ****$p < 0.00001$, 'ns' indicates statistical non-significance. Black asterisks indicate comparison with the control condition of 25°C throughout, genetically unmanipulated. Red asterisks indicate comparisons with control genotype exposed to 32°C heat stress during the embryonic critical period. **(B)** Heat stress experienced during the embryonic critical period (32°C vs. 25°C control) leads to increased presynaptic active zone number. Simultaneous genetic manipulation of muscle DA1 during embryonic stages suggests that ROS by Complex-I is necessary and sufficient to induce active zones changes presynaptically. "Control" indicates control genotype heterozygous for Oregon-R and *DA1-GAL4*. Larvae were reared at the control temperature of 25°C until the late wandering stage, 100 h after larval hatching (ALH). Data are shown with mean ± SEM, ANOVA, ****$p < 0.00001$, 'ns' indicates statistical non-significance. Black asterisks indicate comparison with the control condition of 25°C throughout, genetically unmanipulated. Red asterisks indicate comparisons with control genotype exposed to 32°C heat stress during the embryonic critical period. See raw data in S7 Data.
(TIFF)

**S3 Fig. Characterization of aCC-GAL4 [2] expression.** Expression starts around 13 h after egg lay (AEL) in few sporadic cells, not including the aCC motoneuron and mostly limited to the head region. First aCC motoneurons show GAL4 expression as of 14.5 h AEL, with more aCC motoneurons expressing from 15 h AEL onwards (i.e., 2 h prior to critical period opening). Concomitantly, GAL4 expression in other cells disappears. From laraval hatching onwards, segmentally repeated expression in all aCC motoneurons is maintained until at least 72 h after larval hatching (mid-3rd instar stage). Scale bar = 100 μm.
(TIFF)

**S4 Fig. RET-ROS and HIF-1α signaling has lasting impact on NMJ only during the critical period. (A)** Experimental paradigm. **(B)** Genetic manipulation of motoneuron aCC during the critical period *versus* after critical period closure. "Control" indicates control genotype heterozygous for Oregon-R and *aCC-GAL4[2]*; *Auxin-GAL80*. Larvae were reared at the control temperature of 25°C until the late wandering stage, 100 h ALH. Scale bar = 20 μm. **(C)** Dot-plot quantification shows changes to aCC NMJ growth on its target muscle DA1, based on the standard measure of the number of boutons (swellings containing multiple presynaptic release sites/active zones). Data are shown with mean ± SEM, ANOVA, ****$p < 0.00001$, 'ns' indicates statistical non-significance. **(D)** Dot-plot quantification shows changes in the number of active zones at aCC NMJs quantified in C). Data are shown with mean ± SEM, ANOVA, ***$p < 0.0001$, ****$p < 0.00001$, 'ns' indicates statistical non-significance. See raw data in S8 Data.
(TIFF)

**S5 Fig. RET-ROS and HIF-1α signaling have no lasting impact on behavior after the critical period. (A)** Experimental paradigm. **(B)** Crawling speed of third instar larvae (72 h after larval hatching (ALH)). Temperature experienced during the embryonic critical period (25°C control vs. 32°C heat stress) and simultaneous genetic manipulation of all neurons during first instar stage. "Control" indicates control genotype heterozygous for Oregon-R and *CNS-GAL4; Auxin-GAL80*. Larvae were reared at the control temperature of 25°C until 72 h ALH. Each data point represents crawling speed from an individual uninterrupted continuous forward crawl, $n$ = specimen replicate number, up to three crawls assayed for each larva. Mean ± SEM, ANOVA, ns = not significant, ****$p < 0.00001$. See raw in S9 Data.
(TIFF)

**S6 Fig. Homeostatic responses to environmental change are maintained after 32°C critical period. (A)** Acute shift of temperature. Crawling speed of third instar larvae (72 h after larval hatching (ALH)). Temperature experienced during the embryonic critical period (25°C control vs. 32°C heat stress); temperature experienced during the larva rearing (25°C control vs. 32°C heat stress) and control (25°C) versus acute shift of temperature during the crawling assay (32°C). Genotype is Oregon-R. Each data point represents crawling speed from an individual uninterrupted continuous forward crawl, $n$ = specimen replicate number, up to three crawls assayed for each larva. Mean ± SEM, ANOVA, ns = not significant, ****$p$ < 0.00001. **(B)** Chronic adaptation to the heat stress. Crawling speed of third instar larvae (72 h after larval hatching (ALH)). Temperature experienced during the embryonic critical period (25°C control vs. 32°C heat stress); temperature experienced during the larva rearing and during the crawling assay (25°C control vs. 32°C chronic adaptation). Oregon-R. Each data point represents crawling speed from an individual uninterrupted continuous forward crawl, $n$ = specimen replicate number, up to three crawls assayed for each larva. Mean ± SEM, ANOVA, ns = not significant. See raw data in S10 Data.
(TIFF)

**S1 Data. Raw date of results in Fig 1.**
(XLSX)

**S2 Data. Raw date of results in Fig 2.**
(XLSX)

**S3 Data. Raw date of results in Fig 3.**
(XLSX)

**S4 Data. Raw date of results in Fig 4.**
(XLSX)

**S5 Data. Raw date of results in Fig 5.**
(XLSX)

**S6 Data. Raw date of results in S1 Fig.**
(XLSX)

**S7 Data. Raw date of results in S2 Fig.**
(XLSX)

**S8 Data. Raw date of results in S4 Fig.**
(XLSX)

**S9 Data. Raw date of results in S5 Fig.**
(XLSX)

**S10 Data. Raw date of results in S6 Fig.**
(XLSX)

## Acknowledgments

The authors are grateful to Mihaela Serpe for the Rabbit polyclonal anti-GluRIIB antiserum.

The authors are grateful to Tony Southall, Alex Whitworth, Joe Bateman, and Alberto Sanz for generously providing fly stocks. Stocks obtained from the Bloomington Drosophila Stock Center (NIH P40OD018537) were used in this study. The nc82 and 1D4 monoclonal antibodies were obtained from the Developmental Studies Hybridoma Bank, created by the

NICHD of the NIH and maintained at The University of Iowa, Department of Biology, Iowa City, IA 52242, USA. The work benefited from the Imaging Facility, Department of Zoology, supported by Matt Wayland, and funds from a Wellcome Trust Equipment Grant (WT079204) with contributions by the Sir Isaac Newton Trust in Cambridge, including Research Grant [18.07ii(c)]. Work on this project benefited from the Manchester Fly Facility, established through funds from the University of Manchester and the Wellcome Trust (Grant 087742/Z/08/Z).

## Author contributions

**Conceptualization:** Daniel Sobrido-Cameán.

**Funding acquisition:** Richard A. Baines, Matthias Landgraf.

**Investigation:** Daniel Sobrido-Cameán, Bramwell Coulson, Michael Miller, Matthew C. W. Oswald, Tom Pettini, David M. D. Bailey.

**Supervision:** Richard A. Baines, Matthias Landgraf.

**Writing – original draft:** Daniel Sobrido-Cameán, Matthias Landgraf.

**Writing – review & editing:** Daniel Sobrido-Cameán, Bramwell Coulson, Matthew C. W. Oswald, Tom Pettini, Richard A. Baines, Matthias Landgraf.

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
