## [Editor Report · Decision Letter 0]

3 Feb 2025

Dear Dr Sobrid-Camean, 

Thank you for submitting your manuscript entitled "Mitochondrial ROS and HIF-1α signalling mediate critical period plasticity" for consideration as a Research Article by PLOS Biology.

Your manuscript has now been evaluated by the PLOS Biology editorial staff as well as by an academic editor with relevant expertise and I am writing to let you know that we would like to send your submission out for external peer review.

Once your full submission is complete, your paper will undergo a series of checks in preparation for peer review. After your manuscript has passed the checks it will be sent out for review. To provide the metadata for your submission, please Login to Editorial Manager (https://www.editorialmanager.com/pbiology) within two working days, i.e. by Feb 05 2025 11:59PM.

Kind regards,

Christian

Christian Schnell, PhD

Senior Editor

PLOS Biology

cschnell@plos.org

---

## [Decision Letter · Decision Letter 1]

12 Mar 2025

Dear Dr Sobrid-Camean,

Thank you for your patience while your manuscript "Mitochondrial ROS and HIF-1α signalling mediate critical period plasticity" was peer-reviewed at PLOS Biology. It has now been evaluated by the PLOS Biology editors, an Academic Editor with relevant expertise, and by several independent reviewers. 

In light of the reviews, which you will find at the end of this email, we would like to invite you to revise the work to thoroughly address the reviewers' reports.

As you will see below, the reviewers agree that the study is overall well done and provides important insights. Reviewer 2 has only a few minor suggestions, but Reviewer 1 and Reviewer 3 request a number of additional control experiments and analyses that will be required to fully address their concerns.

Given the extent of revision needed, we cannot make a decision about publication until we have seen the revised manuscript and your response to the reviewers' comments. Your revised manuscript is likely to be sent for further evaluation by all or a subset of the reviewers.

**IMPORTANT - SUBMITTING YOUR REVISION**

*Re-submission Checklist*

*Published Peer Review*

*PLOS Data Policy*

*Blot and Gel Data Policy*

Sincerely,

Christian

Christian Schnell, PhD

Senior Editor

PLOS Biology

cschnell@plos.org

REVIEWS:

Reviewer #1 (James Jepson): Mitochondrial ROS and HIF-1α signalling mediate critical period plasticity. Sobrido-Cameán et al., (2025). 

Overview: 

In this work, Sobrido-Cameán and colleagues investigate the molecular mechanisms underlying a critical period of sensitivity to heat stress during the development of the Drosophila larval locomotor network. 

The authors show that heat stress during a narrow window of developmental time (17-19 h after egg laying) causes long-lasting alterations in pre- and post-synaptic development and neural excitability during the larval stages of the Drosophila lifecycle. Using elegant genetic approaches, they identify mitochondrial reactive oxygen species (ROS) as a key mediator of these alterations. Furthermore, they provide evidence that the conserved epigenetic regulator HIF-1alpha/Sima acts downstream of ROS to drive these long-term changes in development and intrinsic excitability. 

The identification of ROS and HIF-1alpha as effectors of a critical period influencing locomotion is both novel and exciting, and may help further elucidate the molecular underpinnings of critical periods during neural development. Since these developmental phases are disrupted in a range of neurological disorders, this work both advances the fundamental understanding of neural development and has potential clinical relevance. 

The studies themselves are well performed and include technically challenging electrophysiological recordings from larval motoneurons. There are certain controls that could be added to improve robustness. Additional analyses of existing data and textual alterations could provide further mechanistic insights and help the readers understand the work more clearly. Finally, it would be highly beneficial if the authors could directly test for increases in pre- and post-synaptic ROS in response to heat stress, as predicted by their model and prior data. With these inclusions, I would be very happy to see this interesting work published in PLoS Biology. 

Major comments:

1. ROS levels. 

The authors use several independent genetic manipulations to show that ROS is required post-synaptically for heat-induced changes in post-synaptic GluR11A levels, and presynaptic bouton number (Figure 1), pre-synaptically for heat-induced changes in presynaptic bouton number (Figure 3), and pre-synaptically for heat-induced changes in intrinsic motoneuron excitability (Figure 5). The authors mention prior work showing that ROS can mediate activity-dependent synaptic growth (a presynaptic effect), and that heat stress elevates ROS (page 3). However, they do not directly show that in their experimental paradigm, embryonic heat stress increases ROS, and importantly, whether this occur both pre- and post-synaptically (as predicted). 

I agree with the authors that this is likely to occur, but if it did not, it would significantly alter their mechanistic model. Therefore, it would be beneficial for the authors to test whether ROS is indeed elevated by heat stress during the embryonic critical period, in motoneurons and the post-synaptic muscle tissue, perhaps using transgenic reporters such as UAS-mito-roGFP2-Orp1, as they have used previously (Oswald et al., (2018) eLife). 

2. Methods for transient transgene expression.

The schematics in Figures 1-6 depict transient control of transgene expression during (approximately) the embryonic critical period. How this is achieved is not immediately clear to the reader - the AGES system is only described after Fig. 3B-D are mentioned (for Supp. Fig. 1), and in the legend of Fig. 5. Can the authors please clearly explain to the reader how they control transgene activity at the earliest stage (i.e for Figure 1), and mention the logic of using the AGES system? (I presume this was used because the more commonly used TARGET system requires temperature changes?). 

This leads to an important control condition that appears to be currently missing in some datasets - an auxin-fed transgene or driver-alone control containing auxin-Gal80. This genotype appears present in Figure 5 and Supp. Figs 1 and 2, but it was not clear to me whether this control was also fed auxin at the same time as the experimental genotypes (apologies if I missed it). Please clarify in the text and add this control to relevant datasets if absent. 

3. RNAi line controls.

The authors use various RNAi lines (#26207, 50727, 34717) that are in a y, sc, v, sev quadruple mutant background (Figures 4-5). Have the authors tested RNAi alone controls to confirm that synaptic development is not altered in male hemizygotes or heterozygote females for these mutants? 

4. Morphological analyses at the NMJ.

At the larval NMJ, the authors focus on the effect of heat stress on presynaptic bouton number and post-synaptic GluR expression. The images provided by the authors generated questions to me about two other aspects of NMJ development. Firstly, whether heat stress also affects bouton size? This seems to decrease somewhat in the 32C control, but unlike bouton number, this property does not seem altered by AOX over-expression (at least, in the image provided; Fig. 1B). This raises the question of whether some morphological alterations are driven by ROS, but others are ROS-independent? Secondly, have the authors quantified ghost boutons at the NMJ (PMID: 18341991), and tested whether their number is affected by heat stress? Again, images in Fig. 1B are suggestive of potential alterations. A decrease in ghost bouton number might suggest a pathway to increase mature bouton number. 

These suggestions are not essential to address but could yield interesting mechanistic insights. 

5. Presynaptic and postsynaptic effects of ROS/HIF-1alpha. 

The authors identify a mix of pre- and post-synaptic requirements for ROS and HIF-1alpha in driving heat-induced changes in development during the critical period. ROS are required post-synaptically for changes in bouton number and GluRIIA levels Fig.1), and pre-synaptically for increased bouton and active zone number (Fig.3) and reduced intrinsic excitability (Fig.5); while HIF1-alpha is required post-synaptically for changes in bouton number and GluRIIA levels (Fig.4), and pre-synaptically for reduced intrinsic excitability (Fig.5). 

These complex requirements emphasize the importance of determining where ROS is increasing (see above) in response to heat stress. The results also suggest that post-synaptic ROS/HIF1-alpha induce retrograde signals that impact presynaptic bouton number. However, it is unclear whether the reverse is true: can enhancing presynaptic ROS/HIF1-alpha act trans-synaptically to alter post-synaptic GluRIIA levels? 

In the context of temperature-induced changes in larval crawling, the above results also made we wonder whether the authors had tested how muscle-specific manipulations of AOX/HIF1 affected crawling speed; and/or whether the authors had distinguished whether this pathway was relevant solely in motoneurons (using a motoneuron-specific driver) for adaptations to heat stress, or were required more broadly throughout the CNS? 

Again, these are not essential experiments but their inclusion might help more narrowly define relevant domains in which the ROS/HIF1 axis is important during the critical period. 

6. Basis of reduced intrinsic excitability (Figure 5). 

The authors show that heat stress during the critical period reduced AP firing rate without altering RMP. How do the authors stipulate that AP frequency is being altered? Could the authors examine AP shapes in their datasets to test whether this property is affected by embryonic heat stress? For example, could the duration/amplitude of the afterhyperpolarization current be increased? Or spike width?

Minor comments/questions:

1. How do the authors know that feeding auxin to gravid females induces transgene expression during the latter half of the embryonic stage? (Supp. Fig. 1A). 

2. Descriptions of manipulations. E.g Fig.5, p.6. For all figures, can the authors please specific in the text whether transgene manipulation is occurring pre- or post-synaptically? i.e the effect is cell-autonomous or non-cell-autonomous. 

3. Figure 5: it was unclear to me why HIF-1alpha and VHL manipulations are combined here (Fig.5C-D) but not elsewhere in the text. Can the authors please clarify and provide rationale? 

4. Please define the 'CNS-Gal4' - I presume this is R57C10-Gal4 (Table 1), but I could not find a statement to this effect. 

5. Discussion: the authors state they 'demonstrate the cell-autonomous necessity of RET-generated mitochondrial ROS and HIF-1a signalling for heat stress-induced critical period plasticity' (p.8). However, many effects described in this manuscript are non-cell-autonomous (e.g retrograde changes at the NMJ). Can the authors clarify that some of the effects are cell-autonomous, whereas others are non-cell-autonomous? 

Reviewer #2: Sobrido-Cameán et al. demonstrate that temperature stress during the late embryonic critical period (CP) induces synaptic plasticity at the Drosophila neuromuscular junction. Using mutant lines and analyses of presynaptic morphology, glutamate receptor expression, synaptic active zones, electrophysiology, and crawling behavior, they reveal that CP plasticity is driven by ROS generation via reverse electron transfer, leading to HIF1-alpha induction.

Comments:

- The legend of figure 4 is missing and seems to be copy-pasted from figure 2.

- A recent study (PMID: 39721493) showed onset of presynaptic plasticity after acute emission of mitochondrial hydrogen peroxide in late-stage larvae. The authors should cite this work and discuss how it relates to their finding that RET-ROS induces plasticity if induced during but not after the critical period (supp fig 1). 

Reviewer #3: The importance of critical periods during neurodevelopment cannot be understated; disruption of synaptic activity during developmental critical periods has lasting effects on synaptic morphology, growth, function, and behavioral coordination. Though the existence of critical periods has long been known, the events that occur during those critical periods and importantly, the molecules needed to enact them have remained considerably more mysterious. Sobrido-Cameán and colleagues here identify two fundamental new concepts regarding critical period biology: first, that manipulating a single cell during a critical period can influence later changes in morphology, protein recruitment, and activity; and second, that reactive oxygen species (ROS) from mitochondria and acting through HIF-1α are required to mediate critical period signals. Mitochondrial ROS is required to signal in postsynaptic muscles or presynaptic motoneurons - in the absence of ROS (either through failure to generate or subsequent degradation), critical period changes induced by high temperature bouts are blocked. These changes are seen morphologically (involving bouton number, GluRIIA abundance, or active zone number) and functionally (involving action potential number or crawling ability. In muscles, blocking (either by directly impairing or expressing enzymes that regulate its levels) HIF-1α precludes ability to induce changes during the critical period. 

The findings represent a significant advance in our understanding of the molecular basis of critical period signaling. Moreover, the data posit a fascinating new addition to the burgeoning concept that ROS are not just a deleterious byproduct of existence, but rather, an important neuronal signaling axis. The experiments are particularly well designed and the genetics very well controlled, with examples of loss-of-function and gain-of-function consistent with each other. Moreover, the use of exogenously applied constructs (like yeast Ndi1 and AOX) to demonstrate similar manipulations of ROS is particularly helpful. There are, however, a few points that should be addressed in a revised manuscript.

Major Issues:

1) Does this work transsynaptically? When mitochondrial ROS is manipulated in motoneurons, does this influence GluRIIA abundance? When mitochondrial ROS or HIF-1α is manipulated in muscle, does this influence active zones? Similarly (if possible), does concurrent transsynaptic manipulation enhance the effects during the critical period or does one cell basically hit the ceiling of how much the system can change?

2) The main figures focus on HIF-1α in muscles with some evidence from the supplement that HIF-1α may also function in motoneurons? Can this data be elevated from the supplement to the main paper and also expanded upon? It is helpful to know if the same downstream effector occurs pre- and postsynaptically.

3) Specifically regarding the change in active zone number: do ROS manipulations alter active zone density? The changes show an increase in bouton number, so naturally, it would be expected if active zone number scales with bouton number (as has been shown developmentally), then this would naturally occur. Therefore, this is expected - unless there is some shift in active zone density, which should be examined.

4) When Ndi1 is expressed in DA1 at 25 degrees, there is an induction of synaptic changes during the critical period? If this manipulation is done at 32 degrees, do the two enhance each other? Or do they appear similar to a single manipulation? This would be helpful in ensuring that (at least genetically), they function in the same oeuvre. 

5) The authors use multiple methods to regulate HIF-1α levels. Are there suitable reagents to examine HIF-1a levels to validate these methods in cells examined?

Minor Issues:

1) Though I understand the concept of Figure 5B, it would be very challenging for a non-physiologist to see the phenotype the authors are describing? Could you consider a different way of presenting? Perhaps picking one specific current and showing the results? Are the changes significant? Do they trend? 

2) Though Table 1 is helpful, strain source data should be in the first Materials and Methods section. It would be much easier that way to determine provenance of the reagents used.

---

## [Decision Letter · Decision Letter 2]

21 Jul 2025

Dear Daniel,

Thank you for your patience while we considered your revised manuscript "Mitochondrial ROS and HIF-1α signalling mediate critical period plasticity" for publication as a Research Article at PLOS Biology. This revised version of your manuscript has been evaluated by the PLOS Biology editors, the Academic Editor and two of the original reviewers.

Based on the reviews and on our Academic Editor's assessment of your revision, we are likely to accept this manuscript for publication, provided you satisfactorily address the remaining points raised by the reviewers. Please also make sure to address the following data and other policy-related requests:

* We would like to suggest a different title to improve its accessibility for our broad audience: "Mitochondrial ROS and HIF-1α signaling mediate synaptic plasticity in the critical period"

* Please add the links to the funding agencies in the Financial Disclosure statement in the manuscript details.

* DATA POLICY:

Regardless of the method selected, please ensure that you provide the individual numerical values that underlie the summary data displayed in the following figure panels as they are essential for readers to assess your analysis and to reproduce it: 1BDEF, 2CDE, 3BDEFG, 4DE, 5DEGHIJK, S1AB, S2BD, S4CD, S5B and S6AB.

* CODE POLICY

We expect to receive your revised manuscript within two weeks. 

*Published Peer Review History*

*Press*

Sincerely,

Christian

Christian Schnell, PhD

Senior Editor

cschnell@plos.org

PLOS Biology

Reviewer remarks:

Reviewer #1: The authors have fully addressed my initial comments. In their revised text, they have included both new data and further analyses of thier original datasets that have collectively enhanced the robustness of their findings and conclusions. Of particular note, they have generated and utilised a novel transgenic reporter of ROS to demonstrate that increasing ambient temperature during the embryonic critical period indeed enhances ROS levels both pre- and post-synaptically - a finding that adds support to their mechanistic model. They have also modified the text to further enhance the clarity of the manuscript. I congratulate the authors on their excellent work, which has revealed exciting new insights into the molecular basis of critical periods influencing motor circuit development. I look forward to seeing their manuscript in press.

Reviewer #3: The authors have done a commendable job addressing reviewer comments and I applaud them for the careful thought, rigorous consideration, and the additional new data provided. I think this is suitable for publication save one minor adjustment:

1) I appreciate the addition of DA1 muscle and aCC neuron diagrams but their current iteration is far too small for them to have any true added benefit to the figures. Can these diagrams be made much bigger so that the reader can understand better the types of experiments being done?

---

## [Editor Report · Decision Letter 3]

30 Jul 2025

Dear Daniel,

Thank you for the submission of your revised Research Article "Mitochondrial ROS and HIF-1α signaling mediate synaptic plasticity in the critical period" for publication in PLOS Biology. On behalf of my colleagues and the Academic Editor, Timothy Mosca, I am pleased to say that we can in principle accept your manuscript for publication, provided you address any remaining formatting and reporting issues. These will be detailed in an email you should receive within 2-3 business days from our colleagues in the journal operations team; no action is required from you until then. Please note that we will not be able to formally accept your manuscript and schedule it for publication until you have completed any requested changes.

PRESS

Sincerely, 

Christian

Christian Schnell, PhD

Senior Editor

PLOS Biology

cschnell@plos.org